# MULTI-PLAYER MULTI-ARMED BANDITS WITH DELAYED FEEDBACK

## ABSTRACT

Multi-player multi-armed bandits have been researched for a long time due to their application in cognitive radio networks. In this setting, multiple players select arms at each time and instantly receive the feedback. Most research on this problem focuses on the content of the immediate feedback, whether it includes both the reward and collision information or the reward alone. However, delay is common in cognitive networks when users perform spectrum sensing. In this paper, we design an algorithm DDSE (Decentralized Delayed Successive Elimination) in multi-player multi-armed bandits with stochastic delay feedback and establish a regret bound. Compared with existing algorithms that fail to address this problem, our algorithm enables players to adapt to delayed feedback and avoid collision. We also derive a lower bound in centralized setting to prove the algorithm achieves near-optimal. Numerical experiments on both synthetic and real-world datasets validate the effectiveness of our algorithm.

## 1 INTRODUCTION

Multi-armed Bandits (MAB) is a classic framework widely applied in diverse fields such as online advertising, clinical trials, and recommendation systems. In this framework, a single player sequentially selects an arm $k$ from a finite set $[K] := \{1, ..., K\}$ and receives a random reward $X_k(t)$. However, in many real-world scenarios, the standard MAB framework may not adequately capture the complexities involved. Considering cognitive radio systems which aim that spectrum resources are shared efficiently to users, a key difference from the traditional MAB problem is that when users select the same channel, they collide and no message is transmitted. This situation motivates multi-player multi-armed bandits (MMAB) framework in which $M$ players simultaneously pull arms. If two or more players pull the same arm, their rewards turn to zero which represents failed transmission.

In multi-player bandits, the problem is categorized into centralized and decentralized settings. In the centralized setting, players can freely share their rewards without any loss. Whereas this direct communication would consume substantial energy in cognitive networks, recent studies have primarily focused on the decentralized problem, where players cannot communicate directly. This setting is more complex than centralized MMAB because it requires additional techniques to simulate communication between players. Most recent studies on decentralized MMAB (Boursier & Perchet, 2019; Wang et al., 2020) simulate communication between players by forcing collisions, as the occurrence or absence of a collision provides binary information on optimal arms.

However, in practical cognitive radio networks, a more realistic scenario involves users experiencing delays in signal reception due to various inherent factors. These delays arise from spectrum analysis, where different link layer protocols are needed for different spectrum bands to handle path loss and wireless link errors, leading to different packet transmission delays at the link layer (Akyildiz et al., 2006; Ahmad et al., 2020). Although these delays are common in real-world cognitive radio networks, current research on decentralized MMAB (Xiong & Li, 2023; Xu et al., 2023; Richard et al., 2024) largely overlooks this issue and most existing works discuss the setting that rewards are immediately revealed after players pull arms. Actually, this setting does not align with the practical challenges faced by users, where delays significantly alter the effectiveness of algorithms.

Delayed feedback in single-player bandits has received much attention for several years (Joulani et al., 2013; Lancewicki et al., 2021; Tang et al., 2024). In their model, a player selects an arm but

Table 1: Comparison of lower bound and upper bounds of algorithms. The first row comes from Theorem 1. The second row is derived from Corollary 1, the third row is based on Theorem 2, and the last row comes from Theorem 3. Define $\tilde{d}_1 := \mathbb{E}[d] - \sqrt{\sigma_d^2 \theta/(1-\theta)}$, $\tilde{d}_2 := \mathbb{E}[d] + \sqrt{\sigma_d^2 \log(1/(1-\theta))}$ and $\tilde{d}_3 := \mathbb{E}[d] + \sqrt{\sigma_d^2 \log(K)}$, where $\theta \in (0,1]$ is a quantile of delay distribution. $\sigma_d^2$ is the sub-Gaussian parameter of delay distribution and $\mathbb{E}[d]$ is the expectation. We also define $\Delta_k := \mu_{(M)} - \mu_{(k)}$.

| Setting | Algorithm | Regret bound |
| --- | --- | --- |
| Centralized lower bound | | $\Omega\left(\sum_{k>M} \frac{\log(T)}{\theta \Delta_k} + \frac{M \sum_{k>M} \Delta_k}{K} \tilde{d}_1 - \frac{2}{\theta}\right)$ |
| Centralized | DDSE | $O\left(\sum_{k>M} \frac{\log(T)}{\theta \Delta_k} + \tilde{d}_2 + \frac{M \sum_{k>M} \Delta_k}{K-M} \mathbb{E}[d]\right)$ |
| Decentralized | DDSE | $O\left(\sum_{k>M} \frac{\log(T)}{\theta \Delta_k} + \tilde{d}_2 + \frac{M \sum_{k>M} \Delta_k}{K-M} \tilde{d}_3\right)$ |
| Decentralized | DDSE (simplified)[1] | $O\left(\sum_{k>M} \frac{\log(T)}{\theta \Delta_k} + \frac{\tilde{d}_2 \tilde{d}_3}{KM} + \frac{\tilde{d}_3}{\theta KM \sum_{k>M} \Delta_k^2} + \exp(\frac{\mathbb{E}[d]}{KM} + \frac{\sigma_d^2}{K^2 M^2})\right)$ |

observes the reward only after a period of delay. Centralized MMAB can be tackled by slightly adjusting the well-studied single-player bandit algorithms because players know the exploration results of others at each time. In contrast, the decentralized problem is more difficult. Players have to simulate communication by sending collisions but the feedback of collisions is delayed as well. More importantly, since players are independent and do not know others, straightforward applications of single-player algorithms do not work because players will all attempt to sample the same best arm.

Current algorithms for decentralized MMAB, which rely on immediate feedback to coordinate player actions, are ill-suited to scenarios where delays are introduced. These algorithms typically depend on the timely reception of collision feedback to allow players to adjust their policies and avoid future collisions. However, when feedback is delayed, players cannot determine the success or failure of their actions in real time. This leads to a breakdown in the coordination among players, resulting in frequent collisions and inefficient exploration of the arms. Therefore, existing decentralized MMAB algorithms are not equipped to handle the complexities introduced by delayed feedback, necessitating the development of new algorithms that address these challenges.

## 1.1 Contribution

Motivated by the pressing challenge of delay in cognitive radio networks, we propose a novel bandit framework where multiple players engage in a multi-armed bandit and if two or more players select the same arm, none of them receive the reward. Crucially, in our framework, players receive feedback after a period of stochastic delay, which complicates their ability to learn and adapt in real time, making it exceedingly difficult to avoid collisions and optimize performance.

For this problem, we introduce an algorithm DDSE (Decentralized Delayed Successive Elimination ), where players are divided into a leader and several followers. The leader explores all arms and gradually eliminates sub-optimal arms, while followers pull arms only from the set of best empirical arms. Before each exploration phase begins, players coordinate to use the same best empirical arm set based on the estimation of delay, ensuring that no collision occurs. At regular intervals, the leader communicates the update to followers also using the coordinated set so that followers stay synchronized and receive correct information.

Table 1 compares the regret bound of our algorithm with DDSE_without_delay_estimation which is a simplified version of DDSE. In this version, players do not make estimations on delay and directly pull arms in the latest updated set of best empirical arms. This leads to collisions after every communication ends and derives $O(\tilde{d}_3/\theta KM \sum_{k>M} \Delta_k^2) + O(\tilde{d}_2 \tilde{d}_3/KM)$. The regret due to incorrect communication is bounded by $\exp(\mathbb{E}[d]/KM + \sigma_d^2/K^2 M^2)$ which grows exponentially with increasing $\mathbb{E}[d]$ and $\sigma_d^2$.

---

[1]Simplified version of DDSE. In this algorithm, players do not estimate delay and wait for others.

Through careful algorithm design, DDSE successfully performs communication and thus prevents this exponential term. The added term $O(\frac{M\sum_{k>M}\Delta_k}{K-M}\tilde{d}_3)$ is the regret that players coordinate with each other to select the same set of best empirical arms. Compared with $O(\frac{M\sum_{k>M}\Delta_k}{K-M}\mathbb{E}[d])$ in the centralized upper bound, the regret of our algorithm in the decentralized setting differs by only $O(\frac{M\sum_{k>M}\Delta_k}{K-M}\sqrt{\sigma_d^2\log(K)})$, which diminishes when the delay remains stable. Additionally, we establish a lower bound in Table 1 for centralized MMAB with delay, demonstrating that our regret bound is near-optimal.

## 2 PRELIMINARIES

In this section, we describe the formulation for multi-player multi-armed bandits with delayed feedback. For a positive integer $n$, we will use $[n]$ to represent $\{1, 2, ..., n\}$.

Denote $M$ as the number of players and $K$ as the number of arms. Note that $M \leq K$ so that there is at least one arm available for each player without mandatory overlap or collision. At time $s \in [T]$, player $j$ selects an arm $k$ and gains a random reward $X_k^j(s)$ which is drawn i.i.d. according to unknown fixed distribution with expectation $\mu_k \in [0,1]$. Denote $\pi_s^j$ as the arm that is selected by player $j$ at $s$. After pulling their arms, players do not observe feedback immediately. On the contrary, they receive the feedback after delayed $d_s^j$ at $t$, i.e. $s + d_s^j = t$. If more than one players select the same arm, they will collide with each other and none of them gets a reward. We define $\eta_k(s) := \mathbb{1}\{\#C_k(s) > 1\}$ as the collision indicator where $C_k(s) := \{j \in [M] \mid \pi_s^j = k\}$ is the set of players who pull the same arms at time step $s$. Then we define $r^j(s) := X_k^j(s)[1 - \eta_k(s)]$ as the reward that player $j$ selects a arm $k$ at $s$ time step.

In this paper, we discuss collision sensing in which player $j$ receives a tuple $< r_k^j(s), \eta_k^j(s), s >$ where $s$ is the previous time index. In real-world networks, transmission delays are naturally bounded by physical and protocol limits, preventing extreme values (Azarfar et al., 2015). Similarly, in single-player bandits, many works assume that delays are bounded by $d_{\max}$ which is a fixed constant (Li & Guo, 2023; van der Hoeven et al., 2023; Wang et al., 2024). However, we do not adopt this assumption; instead, we introduce a more relaxed assumption that allows for larger delays, but with a low probability of occurrence, which aligns with real-world scenarios where large delays in cognitive networks are rare.

**Assumption 1** *Let $\{d_t^j\}_{t=1,j=1}^{T,M}$ are independent non-negative random variables with sub-Gaussian distribution. Denote $\sigma_d^2$ as the sub-Gaussian parameter and $\mathbb{E}[d]$ as the expectation of the distribution. Then for any $a > 0$,*

$$P(|d_t^j - \mathbb{E}[d]| \geq a) \leq 2\exp(-\frac{a^2}{2\sigma_d^2}).$$

This assumption allows for a practical modeling of the delay without imposing overly restrictive conditions on its behavior, making it reasonable to capture the inherent variability and uncertainty in network delays. We also define $d(\theta) := \min\{\gamma \in \mathbb{N}|P(d \leq \gamma) \geq \theta\}$ as the quantile function of the delay distribution. Note that we allow $\mathbb{E}[d]$ and $\sigma_d$ to be unknown. Then the expected regret is defined as

$$R_T := T\sum_{j\in[M]}\mu_{(j)} - \mathbb{E}\left[\sum_{t=1}^{T}\sum_{j\in[M]}r^j(t)\right],$$

where $\mu_{(j)}$ is $j$-th order statistics of $\mu$, i.e. $\mu_{(1)} \geq \mu_{(2)} \geq ... \geq \mu_{(K)}$.

## 3 ALGORITHM

The proposed algorithm DDSE (Decentralized Delayed Successive Elimination) is composed of exploration phase, communication phase and exploitation phase. Players are divided into one leader and $M - 1$ followers. Define $p_{\max}$ as the maximum number of communication phases within a given time horizon. We also define $\mathcal{M}_p^j$ as the best empirical arm set of player $j$ that the leader

intends to pass to the followers during the $p$-th communication phase. Due to delays, players might have different perceptions of $\mathcal{M}_p^j$, which will lead to collisions. A natural idea is that players use previous $\mathcal{M}_{p'}^j$ where $p' \leq p \leq p_{\max}$ to maintain consistency with others. Considering cognitive wireless sensor networks, sensor nodes are usually pre-deployed (Joshi et al., 2013), so they are equipped with information on the total number of nodes and their ID. Consequently, we assume that each player in our algorithms is initialized with her rank among all players and is aware of the total number of them. Algorithm 1 describes DDSE from the view of the leader. The algorithm from the view of followers is in Appendix C.1.

---

**Algorithm 1** DDSE (Leader)

---
    **Input:** $K$, $M$;
1: Initialize $\mathcal{M}_0^M$ randomly, $\mathcal{K} = [K]$, $e_M = 0$ (ending signal), $p = 0$, $q_M = 0$, $S_k(t) = 0$, $\hat{\mu}_k(t) = 0$, $\hat{\mu}_d^M = 0$ and $(\hat{\sigma}_d^2)^M = 0$;
2: **while** $t \leq T$ **do**
3:      Explore in $\mathcal{M}_{p-q_M}^M$ and $\mathcal{K}/\mathcal{M}_{p-q_M}^M$;
4:      Update $\hat{\mu}_d^M, (\hat{\sigma}_d^2)^M, S_k(t), \hat{\mu}_k(t)$;
5:      Remove from $\mathcal{K}$ all arms $k$ s.t. $|\{i \in \mathcal{K} | LCB_t(i) \geq UCB_t(k)\}| \geq M$;
6:      **if** $t \mod (KM\lceil \log(T) \rceil) = 0$ **then**
7:          $p \leftarrow \frac{t}{KM\lceil \log(T) \rceil}$;
8:          **if** $\mathcal{M}_p^M \neq \mathcal{M}_{p-1}^M$ **then**          ▷ communication phase
9:              Communication$(a_p^-, a_p^+, i_{a_p^-}, e_M, \mathcal{M}_{p-q_M}^M)$;
10:          **else** VirtualCommunication$(\mathcal{M}_{p-q_M}^M)$;
11:          **end if**
12:          Find $q_M$ s.t. (1);          ▷ coordinate to the same best empirical arm set
13:      **end if**
14:      **if** $|\mathcal{K}| = M$ && $e_M = M$ && $q_M = 0$ **then**          ▷ exploitation phase
15:          Select $\mathcal{M}_{p_{\max}}^M(M)$ until $T$.
16:      **end if**
17: **end while**

---

## 3.1 EXPLORATION

Denote $\hat{\mu}_d^j$ as player $j$'s estimation of $\mathbb{E}[d]$ and $(\hat{\sigma}_d^2)^j$ as the estimation of $\sigma_d^2$ from player $j$. We initialize $\mathcal{M}_0^j$ for each player $j \in [M]$ and assign an ID to them. The player with ID $M$ becomes the leader and others are followers. Define $\mathcal{K}$ as the active arm set and it is initialized as $\{1, ..., K\}$. In the beginning, these followers pull arms from the best empirical arm set in a round-robin way. To avoid collision with followers and ensure sufficient exploration, the leader first pulls arms in the set of best empirical arms with followers. Then she selects other arms in $\mathcal{K}$ in a round-robin way while skipping arms in the best arm set. In other words, the leader constantly explores all arms except what has been eliminated. Players also estimate $\hat{\mu}_d^j$ and $(\hat{\sigma}_d^2)^j$ when they receive the feedback.

We define $N_t(k) := \sum_{s<t} \mathbb{1}\{\pi_s^j = k, j = M\}$ as the number of times that the leader chooses arm $k$ before $t$. Define $n_t(k) := \sum_{s<t} \mathbb{1}\{\pi_s^j = k, d_s^j + s < t, j = M\}$ as the number of received feedback of the leader from arm $k$ before $t$. When the leader receives the feedback of arm $k$ at $t$ in exploration phase, she updates

$$UCB_t(k) := \hat{\mu}_k(t) + \sqrt{\frac{2\log(T)}{n_t(k)}}, \ LCB_t(k) := \hat{\mu}_k(t) - \sqrt{\frac{2\log(T)}{n_t(k)}},$$

where $\hat{\mu}_k(t) := \frac{S_k(t)}{n_t(k)}$ is the empirical reward of arm $k$ and $S_k(t)$ is the sum of rewards that the leader has collected on arm $k$ by the end of time $t$. During the exploration phase, she eliminates an arm $k$ from $\mathcal{K}$ at $t$ if there exist more than $M$ arms whose lower confidence bounds are bigger than $UCB_k(t)$.

Due to the influence of delay, $\mathcal{M}_p^j \neq \mathcal{M}_p^l$ if player $l$ does not receive the feedback from the $p$-th communication phase. When they select arms in a round-robin way, different sets of best empirical

arms might lead to collisions. To avoid this situation, players need to select a previous best empirical arm set based on $\hat{\mu}_d^j$ and $(\sigma_d^2)^j$. Specifically, by delay's sub-Gaussian property, we know that when

$$t - pKM \log(T) > \mathbb{E}[d] + \sqrt{2\sigma_d^2 \log(M-1)(K+2M)(T)},$$

$\mathcal{M}_p^j$ from the $p$-th communication phase has been received by all followers with high probability. Therefore, the algorithm aims to identify $q_j \in \mathbb{N}$ which is defined as player $j$'s backward counting number of communication phase, i.e. at the current time step $t$, all players have received results of the $(p - q_j)$-th communication phase, allowing them to use the same $\mathcal{M}_{p-q_j}^j$ to avoid collision caused by delay. Specifically, $q_j$ increases from 0 and when it satisfies

$$t > \hat{\mu}_d^j + \sqrt{2(\hat{\sigma}_d^2)^j \log\left((M-1)(K+2M)(T)\right)} + (p - q_j)KM \log(T), \tag{1}$$

then the $(p - q_j)$-th result of communication is exactly what we want.

Note that the length of each exploration phase is fixed, rather than depending on the number of active arms (Boursier & Perchet, 2019; Wang et al., 2020). This is because players receive feedback at different times and have varying numbers of active arms, making it difficult to maintain synchronization with a dynamic phase length. In our algorithm, players remain synchronized and select arms from the same set of best empirical arms, ensuring collision-free exploration.

Sub-optimal arms in $\mathcal{K}$ are gradually eliminated by the leader. When $|\mathcal{K}| = M$, she waited to communicate with followers about the end of exploration. After that, the leader remains in the exploration phase until $q_M = 0$, at which point she moves to the exploitation phase and pulls $\mathcal{M}_{p_{\max}}^M(M)$. When a follower $j$ receives the ending signal and finds $q_j = 0$, she will enter the exploitation phase and continuously select arm $\mathcal{M}_{p_{\max}}^j(j)$ until $T$.

### 3.2 Communication

Players enter the communication phase every $KM \log(T)$ times and the length of each communication phase is $K + 2M$. On each communication beginning, if $\mathcal{M}_p^M \neq \mathcal{M}_{p-1}^M$, then the leader begins communication. Otherwise, she runs a virtual communication (see Appendix C.1) to maintain synchronization with followers. Motivated by Wang et al. (2020), the communication phase in our algorithm is divided into three parts. The first and second parts are used for removing and adding an arm in $\mathcal{M}_p$. The third part is used for the leader to send the ending signal. Denote $a_p^-$ as the arm to remove and $a_p^+$ as the arm to add in the $p$-th communication phase. We also define $i_k \leq M$ as the position of arm $k \in \mathcal{M}_p^j$.

**Part 1: Remove Arm** The leader firstly identifies $a_p^- \in \mathcal{M}_p^M$ and finds its position $i_{a_p^-}$. Then in this part, the leader selects arm $\mathcal{M}_{p-q}^M(i_{a_p^-})$ for $M$ consecutive rounds. Meanwhile, followers pull arms from $\mathcal{M}_{p-q}^j$ in a round-robin way, ensuring that each follower collides once with the leader. Denote $a_p^c$ as the arm that follower $j$ selects and collides with the leader in the first part of the $p$-th communication. Since $\mathcal{M}_p$ is ordered for all $p \leq p_{\max}$, followers can receive the update of the leader to remove $\mathcal{M}_p^j(i_{a_p^c})$ during the $p$-th communication phase by selecting arms from $\mathcal{M}_{p-q}^j$. Thus, the information is passed successfully even if $\mathcal{M}_p^j$ is incomplete for follower $j$, allowing our algorithm to adapt to large delays.

**Part 2: Add Arm** In this part, the leader continuously pulls $a^+$ for $K$ rounds while followers select arms in $[K]$ in a round-robin way. Each follower also collides once with the leader. After receiving both the collision from Part 1 and Part 2, followers place $a^+$ in the position of $\mathcal{M}_p^j(i_{a_p^c})$, which does not break the order of $\mathcal{M}_p^j$.

**Part 3: Notify End** If $|\mathcal{K}| = M$, it indicates that all the sub-optimal arms have been eliminated and the leader selects arms in $\mathcal{M}_p^M$ sequentially, while followers continuously select arm $\mathcal{M}_p^j(j)$ for $M$ times. Otherwise, the leader does not send collisions by selecting $\mathcal{M}_p^M(M)$ for $M$ times. Finally, each follower receives a collision which means the end of exploration.

**Algorithm 2** Communication (Leader)

**Input:** $a_p^-, a_p^+, i_{a_p^-}, e_M, \mathcal{M}_{p-q_j}^M$;
**Part 1: Remove Arm**
1: **for** $M$ time steps **do**
2:    Select $\mathcal{M}_{p-q_j}^M(i_{a_p^-})$;
3: **end for**
**Part 2: Add Arm**
4: **for** $K$ time steps **do**
5:    Select $a^+$;
6: **end for**
**Part 3: Notify End**
7: **for** $M$ time steps **do**
8:    **if** $|\mathcal{K}| = M$ **then**
9:       $m_i \leftarrow [t \bmod M]$ and $p_{\max} \leftarrow p$;
10:       $e_M \leftarrow e_M + 1$;
11:       Select $\mathcal{M}_{p-q_j}^M(m_i)$;
12:    **else** Select $\mathcal{M}_{p-q_M}^M(M)$. **end if**
13: **end for**

**Algorithm 3** Communication (Follower)

**Input:** $\mathcal{M}_{p-q_j}^j$, $j$ (ID of each player);
**Part 1: Remove Arm**
1: **for** $M$ time steps **do**
2:    $m_i \leftarrow [(t+j) \bmod M]$;
3:    Select $\mathcal{M}_{p-q_j}(m_i)$;
4: **end for**
**Part 2: Add Arm**
5: **for** $K$ time steps **do**
6:    $m_i \leftarrow [(t+j) \bmod K]$;
7:    Select $m_i$;
8: **end for**
**Part 3: Notify End**
9: **for** $M$ time step **do**
10:    Select $\mathcal{M}_{p-q_j}^j(j)$.
11: **end for**

The reason why our communication phase is fixed instead of beginning when $\mathcal{M}_p^j$ changes is that players need to ensure synchronization with others. In Wang et al. (2020), the leader sends a collision to followers as the beginning signal of communication. However, when the feedback of this collision is delayed, followers hardly receive it at the same time and then stagger with the leader. Once players are not aligned with others, followers may receive incorrect information during the communication phase. Furthermore, since communication and exploration are alternating, players might end up selecting the same arm during the exploration phase, resulting in collisions.

Denote $p'$ as the communication phase whose result is the most recent to have been completely received. If the delay is sufficiently small, players can receive the feedback from the $p$-th communication phase before the $(p+1)$-th communication begins. Then $q = 0$ in our algorithm and players continue using $\mathcal{M}_p$. We discuss DDSE_without_delay_estimation which is a simplified version of our algorithm where players directly use $\mathcal{M}_{p'}$ in Appendix C.2. This version does not estimate $\hat{\mu}_d^j$ or $(\hat{\sigma}_d^2)^j$ and also does not coordinate players to pull in the same set of best empirical arms.

## 4 THEORETICAL ANALYSIS

In this section, we present a thorough analysis of our algorithms. The overall regret of multi-player bandits problem is decomposed as $R_T = R_{expl} + R_{com}$, where $R_{expl}$ can be decomposed as $R_{expl} = R_{expl}^L + R_{expl}^F$. We also define $\delta := \min_{1 \le k \le K-1}(\mu_{(k)} - \mu_{(k+1)})$.

### 4.1 CENTRALIZED LOWER BOUND

We first give a lower bound which establishes a foundational standard of this problem. In decentralized multi-player bandits, players intentionally collide with others to simulate communication, which inevitably results in some regret. Therefore, our goal is to minimize the communication duration and the associated regret. To evaluate this, we compare our results with the centralized lower bound to evaluate how the additional information exchange impacts regret reduction.

**Theorem 1** *For any sub-optimal gap set $S_\Delta = \{\Delta_k \mid \Delta_k = \mu_{(M)} - \mu_{(k)} \in [0,1]\}$ of cardinality $K - M$ and a quantile $\theta \in (0,1]$, there exists an instance with an order on $S_\Delta$ and a delay distribution under Assumption 1 such that*

$$R_T \ge \sum_{k>M} \frac{(1-o(1))\log(T)}{2\theta\Delta_k} + \left(\mathbb{E}[d] - \sigma_d\sqrt{\frac{\theta}{1-\theta}}\right)\frac{M}{2K}\sum_{k>M}\Delta_k - \frac{2}{\theta}. \tag{2}$$

The theorem describes the lower bound for centralized multi-player bandits with delayed feedback. The result and demonstrates that our regret bound in Theorem 2 is near-optimal. The full proof of this theorem is provided in Appendix G.

## 4.2 DDSE

**Theorem 2** *In decentralized setting, for delay distribution under Assumption 1, given any $K, M, \mu$ and a quantile $\theta \in (0, 1]$, the regret of DDSE satisfies*

$$R_T \leq \sum_{k>M} \frac{323 \log(T)}{\theta \Delta_k} + \left( 9 + \frac{2M \sum_{k>M} \Delta_k}{K - M} \right) \mathbb{E}[d] + \sigma_d \left( 3\sqrt{6} + 6\sqrt{2 \log(\frac{1}{1-\theta})} \right)$$

$$+ \frac{\sigma_d M}{K - M} \sum_{k>M} \Delta_k \sqrt{\log\left( (M-1)(K+2M) \right)} + C_1,$$

*where $C_1 = \sum_{k>M} \frac{195}{\theta \Delta_k^2} + \frac{4M e^{-\delta^2/2}}{\delta^2}$.*

Compared with Theorem 1, the first term in Theorem 2 is aligned with (2) up to constant factors. The difference between our regret bound and Theorem 1 arises from the decentralized setting, where there is no direct way for players to communicate about rewards and collisions. The regret introduced by the decentralized structure and delay remains independent of $T$. Therefore, our result is near-optimal. Therefore, they need to simulate communication through collisions and wait for other players to maintain a consistent set of the best empirical reward arms. The proof of Theorem 2 is divided into several lemmas and the complete proof can be found in Appendix D.

**Lemma 1** *In decentralized setting, for delay distribution under Assumption 1, given any $K, M, \mu$ and a quantile $\theta \in (0, 1]$, the regret of the exploration phase in DDSE is bounded as*

$$R_{expl} \leq \sum_{k>M} \frac{323 \log(T)}{\theta \Delta_k} + \sum_{k>M} \frac{M \Delta_k}{K - M} \left( 2\mathbb{E}[d] + \sigma_d \sqrt{\log((M-1)(K+2M))} \right)$$

$$+ 3\sigma_d \sqrt{2 \log \left( \frac{1}{1-\theta} \right)} + 3\mathbb{E}[d] + C_2,$$

*where $C_2 = \frac{4M e^{-\delta^2/2}}{\delta^2}$*

This lemma demonstrates that the main regret of DDSE comes from exploration phase. The second term on the right-hand side arises because, after the exploration phase ends, the leader does not begin exploitation immediately. She still needs to select arms from $\mathcal{M}_{p-q_M}^M$ to wait for followers who have not yet received the final feedback. The feedback is delayed for at most $\mathbb{E}[d] + \sigma_d \sqrt{\log((M-1)(K+2M))}$ rounds which is proved in (12), and we multiply it by $\frac{M}{K-M} \sum_{k>M} \Delta_k$. Compared with Theorem 3, we prove that this approach of waiting for others to avoid collisions is much better than ignoring the followers and updating blindly.

**Lemma 2** *In decentralized setting, for delay distribution under Assumption 1, given any $K, M, \mu$ and a $\theta \in (0, 1]$, the regret in the communication phase is bounded by*

$$R_{com} \leq \sum_{k>M} \frac{195}{\theta \Delta_k^2} + 3\sigma_d \left( \sqrt{6} + \sqrt{2 \log(\frac{1}{1-\theta})} \right) + 6\mathbb{E}[d].$$

Players have a communication phase every $KM \log(T)$ rounds, and each communication phase lasts for a fixed duration of $K + 2M$ rounds. Since communication occurs only during the exploration phase, the number of communication phases is $T_{expl}/KM \log(T)$. Therefore, the regret incurred during the communication phases remains constant with respect to $T$.

**Corollary 1** *In centralized setting, for delay distribution under Assumption 1, given any $K, M, \mu$ and a quantile $\theta \in (0, 1]$, the regret of DDSE satisfies*

$$R_T \leq \sum_{k>M} \frac{323 \log(T)}{\theta \Delta_k} + \left( 3 + \frac{M \sum_{k>M} \Delta_k}{K - M} \right) \mathbb{E}[d] + 3\sigma_d \sqrt{\log(\frac{1}{1-\theta})} + C_2,$$

When DDSE runs in centralized setting which means that players can exchange information freely, there is no need for additional communication between players. Followers know the latest exploration results of the leader. Once the leader identifies $\mathcal{M}^*$, they begin exploitation and do not cause regret. Proof of Corollary is in Appendix E.

### 4.3 DDSE WITHOUT DELAY ESTIMATION

**Theorem 3 (Comparison)** *In decentralized setting, for delay distribution under Assumption 1, given any $K, M, \mu$ and a quantile $\theta \in (0, 1]$, the regret of DDSE_without_delay_estimation is bounded by*

$$R_T \leq \sum_{k>M} \frac{323\log(T)}{\theta\Delta_k} + \left(9 + \frac{M\sum_{k>M}\Delta_k}{K-M}\right)\mathbb{E}[d] + \sigma_d\left(3\sqrt{6} + 6\sqrt{2\log(\frac{1}{1-\theta})}\right)$$

$$+ \exp\left(\frac{\mathbb{E}[d]}{KM} + \frac{\sigma_d^2}{2K^2M^2}\right) + O\left(\frac{\tilde{d}_2\tilde{d}_3}{KM} + \frac{\tilde{d}_3}{\theta KM\sum_{k>M}\Delta_k^2}\right) + C_2.$$

If players do not estimate the delay and use the latest best empirical arm set $\mathcal{M}_{p'}^j$, followers will collide with the leader after each communication phase ends. This happens because the leader begins communication after she updates $\mathcal{M}_p^M$, while the followers have not yet received this update, ultimately contributing to a regret of $O(\tilde{d}_2\tilde{d}_3/KM + \tilde{d}_3/\theta KM\sum_{k>M}\Delta_k^2)$. Additionally, note that followers may receive incorrect information during the communication phase if $\mathcal{M}_{p'}^j \neq \mathcal{M}_{p'}^M$, which leads to an exponential regret term $\exp(\mathbb{E}[d]/K + \sigma_d^2/2K^2)$. A more detailed proof is included in Appendix F.

Compare Theorem 3 with Theorem 2 and we find by using $\mathcal{M}_{p-q_j}^j$ instead of $\mathcal{M}_{p'}$, players will not collide with each other after the communication ends, thereby avoiding $O(\tilde{d}_2\tilde{d}_3/KM + \tilde{d}_3/\theta KM\sum_{k>M}\Delta_k^2)$ which could be large when $\Delta_k^2$ is sufficiently small. Moreover, since $\mathcal{M}_{p-q_j}^j = \mathcal{M}_{p-q_l}^l$ for all $j, l \in [M]$, followers receive correct information from the leader, thus eliminating the exponential term in the regret.

## 5 EXPERIMENTS

We conduct various numerical experiments to support our theoretical results. Define $\overline{\Delta} := \sum_{k=1}^{K-1} \frac{\mu_{(k)} - \mu_{(k+1)}}{K-1}$ as the average gap between two consecutive arms in terms of reward. All the results are averaged over 20 runs rounds, with each experiment running for $T = 300,000$ rounds. The default parameters are set as $K = 20$, $M = 10$, $\overline{\Delta} = 0.05$, $\mathbb{E}[d] = 200$ and $\sigma_d = 100$. We consider Gaussian rewards and compare the regret of DDSE with DDSE_without_delay_estimation and SIC-MMAB (Boursier & Perchet, 2019). We also compare with MCTopM, RandomTopM and Selfish in Besson & Kaufmann (2018); Game of Throne in Bistritz & Leshem (2018); ESER in Tibrewal et al. (2019). Parameters are set the same with the original works. The interval and shadow in our figures represent the standard error.

### 5.1 NUMERICAL SIMULATION

To evaluate the performance of our proposed algorithms under varying delay conditions, we conducted two sets of experiments with different delay parameters. In Figure 1, we set $\sigma_d = 50$ and compare the results for different values of $\mathbb{E}[d]$. Each group of four bars with the same color represents the performance of an algorithm under different delay expectations 50, 100, 200, and 500 respectively. Comparison on different $\sigma_d$ is in Appendix B. The experiments show that our algorithms perform significantly better than others. As $\mathbb{E}[d]$ increases, DDSE achieves an improvement of more than twofold in reducing regret compared to DDSE_without_delay_estimation.

Figure 2 reports the performance with varying numbers of players, with DDSE again outperforming other algorithms. We also compare on larger number of players with $M = 30$ and $M = 40$

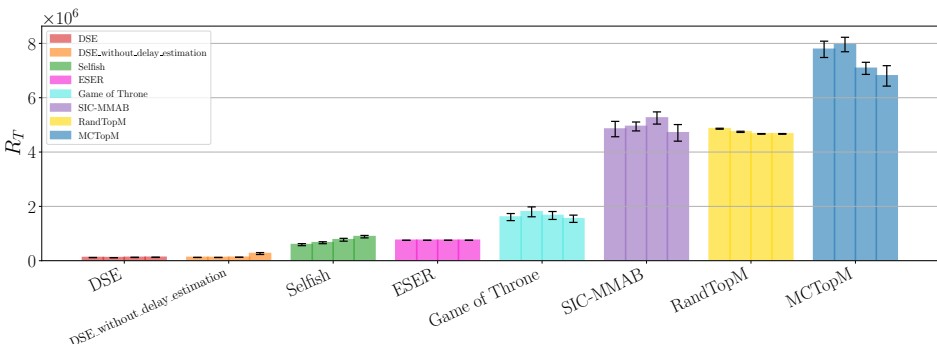

Figure 1: Comparison on $\mathbb{E}[d]$

in Appendix B. The results indicate that when $M$ is small, DDSE_without_delay_estimation performs significantly worse than DDSE. This occurs because the interval of each communication is $KM \log(T)$. When $M$ is small, the interval becomes too short for followers to receive feedback from the most recent communication phase. As a result, followers may obtain incorrect information, leading to staggered exploration, frequent collisions, or premature exploitation. More detailed comparison on $K$ and $\overline{\Delta}$ can be found in Appendix B.

In Figure 2(a), Selfish (Besson & Kaufmann, 2018) also performs well. Besson & Kaufmann (2018) design special UCB index which decreases when collision occurs. However, as the name suggests, players in this algorithm are selfish and only want to maximize their own rewards. Thus, they fail to utilize the exploration results of others, causing the regret to increase rapidly as $M$ grows. Both Game of Throne (Bistritz & Leshem, 2018) and ESER (Tibrewal et al., 2019) follow an explore-then-commit approach, so they rely on the adjustment of parameters heavily. Meanwhile, MCTopM and RandomTopM from Besson & Kaufmann (2018) are built on the Musical Chair framework (Rosenski et al., 2016), where players randomly preempt a chair with no collision. When delay happens, an arm that is identified to be idle in earlier rounds may already have been preempted by other players, but the player always gets out-of-date feedback, resulting in non-stop exploration to find idle arms.

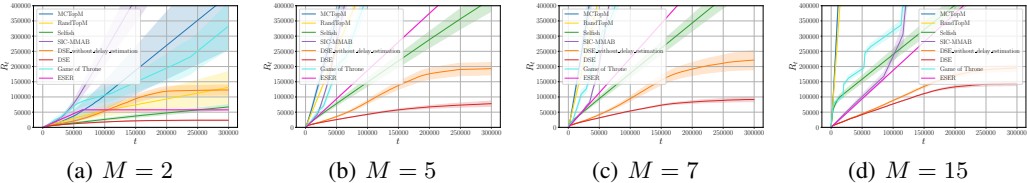

| (a) $M = 2$ | (b) $M = 5$ | (c) $M = 7$ | (d) $M = 15$ |

Figure 2: Comparison between different algorithms on $M$

SIC-MMAB (Boursier & Perchet, 2019) involves communication phase where players exchange rewards with others. However, when feedback is delayed, the communication phase of each player becomes misaligned. While some players find their optimal arms and enter exploitation phase, others remain unaware and continue selecting arms in a round-robin manner. This misalignment leads to collisions with players who have already fixed on their optimal arms.

## 5.2 REAL-WORLD SIMULATION

We evaluate the performance of our algorithms using real-world spectrum measurement data. This dataset[2] was collected in Finland by researchers from the 5G-Xcast project. Figure 3 illustrates a sample of power measurement across four bands in the dataset. Note that in cognitive radio networks, users are divided into preliminary users and secondary users. The aim of cognitive radio networks is that spectrum resources are shared efficiently to secondary users without compromising the critical operations of primary users. Multi-player bandit algorithms are used for secondary users

---

[2]The dataset can be found in https://zenodo.org/records/1293283.

to find available channels. We consider that primarily user signals are on a frequency channel if power measurement is higher than the threshold power level $-90$ dBm, which is the same setting with Alipour-Fanid et al. (2022).

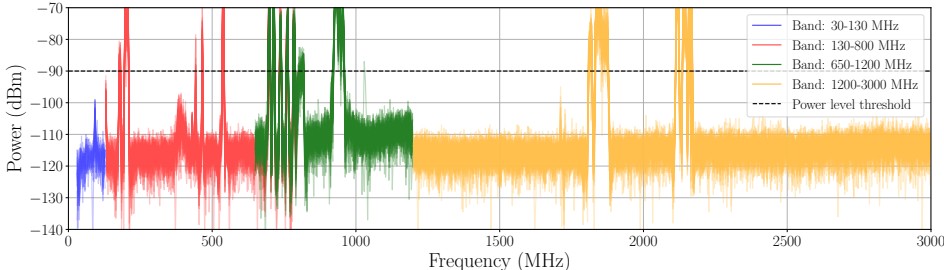

Figure 3: Captured spectrum data from paging frequency bands

Following Wang et al. (2021) and Alipour-Fanid et al. (2022), we consider accumulative throughput and collisions to evaluate the algorithms. The throughput $B$ is computed using Shannon's formula:

$$B = W \log_2(1 + SNR),$$

where $W$ denotes bandwidth and $SNR$ is signal to noise ratio. If the channel is busy (with power bigger than $-90$ dBm), the cognitive radio acquires no throughput, as it enters sleep mode to avoid interfering with primary users. If secondary users select the same channel, the throughput of them is also zero.

Figure 4 illustrates the cumulative throughput over time, highlighting the superior performance of our algorithm. Additionally, Figure 5 compares the cumulative collisions across algorithms. Notably, our algorithm achieves a remarkably low level of cumulative collisions. It is worth mentioning that ESER experiences almost zero collisions due to its mechanism, where players select arms in a round-robin fashion, alternating between exploration and exploitation. In comparison, while our algorithm incurs slightly higher collisions, these are attributed to simulating communication between players. Consequently, our algorithm achieves a lower regret than ESER.

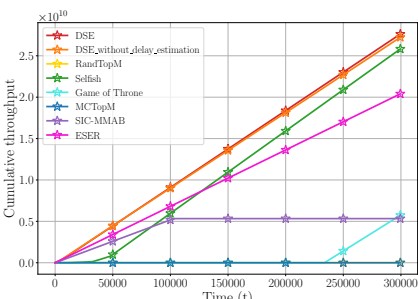

Figure 4: Comparison on throughput

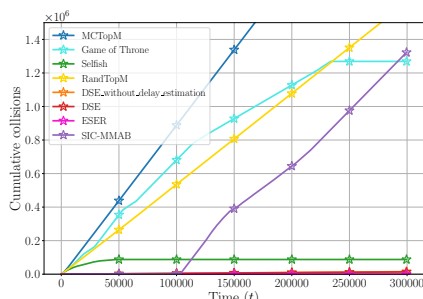

Figure 5: Comparison on collisions

## 6 CONCLUSION

In this paper, we proposed the algorithm DDSE for multi-player multi-armed bandits with delayed feedback. We demonstrated that a decentralized MMAB algorithm can avoid collisions and achieve performance close to its centralized counterpart, even when player feedback is delayed. Rather than allowing players to update blindly, introducing appropriate waiting significantly improves performance and reduces the regret. The lower bound in the centralized setting further confirms that our algorithm is near-optimal. Additionally, practical simulations have validated the superiority of our algorithm. A promising direction for future work would be to study player-dependent delays in multi-player bandits, as delays in cognitive networks often depend on user-specific factors, such as location and device capability.

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

## A    RELATED WORK

The problem of multi-player multi-armed bandits has recently been studied in different settings in the existing literature, where most of the efforts have concentrated on the decentralized setting. Boursier & Perchet (2019) propose an implicit communication mechanism where players intentionally collide to signal information, achieving performance comparable to centralized approaches. Wang et al. (2020) improve this communication phase by electing a leader and only allowing the leader to communicate with followers. Research also has focused on heterogeneous reward settings (Besson & Kaufmann, 2018; Bistritz & Leshem, 2018; Tibrewal et al., 2019; Shi et al., 2021) and adversarial collision scenarios (Mahesh et al., 2022). The challenge of incomplete feedback is another prominent topic (Boursier & Perchet, 2019; Shi et al., 2020; Lugosi & Mehrabian, 2022). Notably, Huang et al. (2022) present near-optimal results under incomplete feedback setting. Wang et al. (2022); Xu et al. (2023) explore the scenario of shareable arms. Recently, Richard et al. (2024) consider asynchronous multi-player bandits in the centralized setting and derive a constant or logarithmic regret.

There has been growing interest in stochastic delay in multi-armed bandits. Vernade et al. (2017) investigate delayed Bernoulli bandits, although their approach requires knowledge of the delay distribution. Pike-Burke et al. (2018) consider scenarios where a sum of observations is received after some stochastic delay. Zhou et al. (2019) explore contextual bandits with stochastic delay. Arm-dependent delay is discussed by Gael et al. (2020), and Lancewicki et al. (2021) later remove the restriction on delay distribution. Tang et al. (2024) focus on strongly reward-dependent delay and achieve near-optimal results. Yang et al. (2024) propose a reduction-based framework to handle delays with sub-exponential distributions.

A similar setting to ours is multi-agent bandits with delay. Existing literature has focused on decentralized cooperative bandits (Cesa-Bianchi et al., 2016; Martínez-Rubio et al., 2019), while non-cooperative game with delay is discussed in Bistritz et al. (2019; 2022). Zhang et al. (2023) consider multi-agent reinforcement learning with both finite and infinite delay. Li & Guo (2023) discuss adversarial bandit problem with delayed feedback from multiple users. Hanna et al. (2024) propose an algorithm in multi-agent bandits with delay and reach a sub-linear regret. However, none of these works consider collisions between players. Since collisions result in a loss of reward, current algorithms in multi-agent bandits cannot be directly applied to our problem.

## B    ADDITIONAL EXPERIMENTS

Figure 6 shows the results for varying values of $\sigma_d$. Each group of four bars in the same color represents the performance of an algorithm under $\sigma_d = 10, 50, 100$, and $150$. The experiments demonstrate that our algorithms significantly outperform others across all settings. As $\mathbb{E}[d]$ increases, DDSE achieves over a twofold reduction in regret compared to DDSE_without_delay_estimation, highlighting its robustness and effectiveness in handling delays.

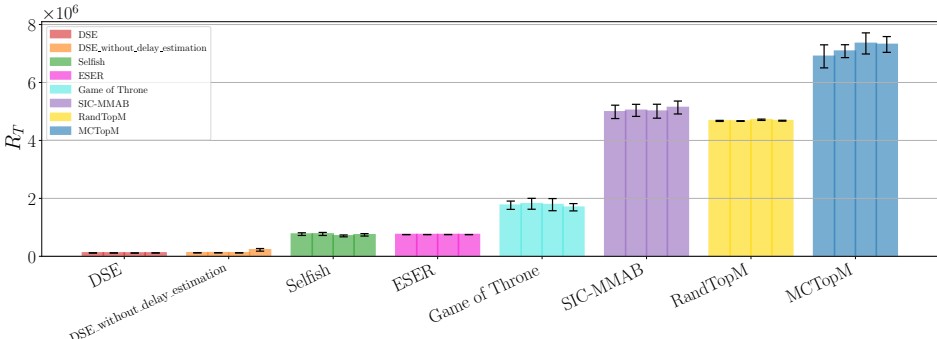

Figure 6: Comparison on $\sigma_d$

Figure 7 reports the performance of various algorithms with larger $M$ and varying numbers of arms. Figure 8 evaluates the impact of $K$ on regret. Among all algorithms, DDSE achieves the best performance except in Figure 8(d). The results indicate that when both $K$ and $M$ are large,

DDSE_without_delay_estimation performs similarly to DDSE. This is because the interval between communication phases is $KM\log(T)$, which is sufficiently long as $K$ and $M$ are big. So players have enough time to receive the results in the last communication phase. However, when $K$ and $M$ are small, the interval becomes too short for followers to receive feedback from the most recent communication phase. As a result, followers may obtain incorrect information, leading to staggered exploration, frequent collisions, or premature exploitation.

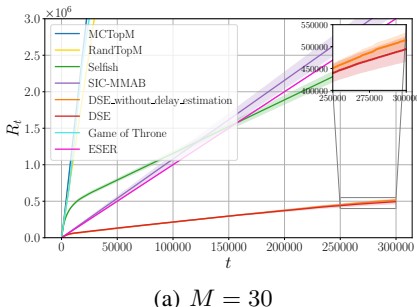 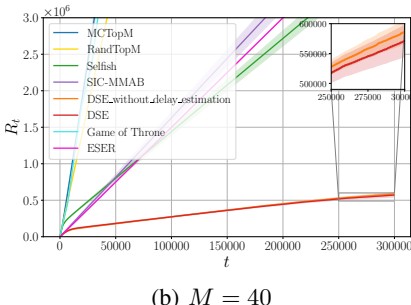

(a) $M = 30$           (b) $M = 40$

Figure 7: Comparison between big $M$

We also note that when $K = 50$ in Figure 8(d), DDSE_without_delay_estimation performs slightly better than DDSE. The reason is that, in DDSE_without_delay_estimation, a large $K$ ensures that followers receive feedback from a communication phase before the next communication phase begins. However, in DDSE, player $j$ adjusts to $\mathcal{M}_{p-q_j}^j$ which is deemed to be received by all players with high probability. This results in the leader being more conservative in exploring sub-optimal arms, causing DDSE to eliminate arms later than DDSE_without_delay_estimation. However, as shown in Figure 9(d), DDSE_without_delay_estimation exhibits significant fluctuations and instability. In cognitive radio systems, where stable signal transmission is desired, DDSE proves to be more robust. It adapts well to different environments and consistently performs effectively, making it a better choice for applications requiring reliable performance.

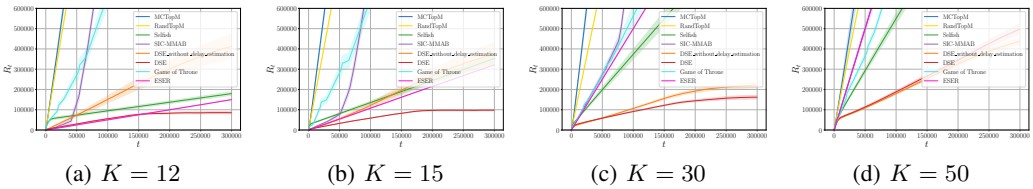

(a) $K = 12$     (b) $K = 15$     (c) $K = 30$     (d) $K = 50$

Figure 8: Comparison between different algorithms on $K$

A detailed comparison of these algorithms is presented in Figure 9, which shows their regrets for different values of $\overline{\Delta}$. As $\overline{\Delta}$ decreases, it becomes harder for the leader to eliminate sub-optimal arms. The reason that SIC-MMAB performs better when $\overline{\Delta}$ is small is that players neither accept nor reject arms within $T = 300,000$ rounds, continuing to select all arms in a round-robin manner. With no changes in $[K]$, collisions are avoided, and the regret does not increase significantly. This also highlights that in multi-player bandits, avoiding collisions is more critical than selecting better arms.

Additionally, as seen in Figure 9(d), DDSE_without_delay_estimation shows large fluctuations. This is because when $\overline{\Delta}$ is small, it becomes difficult for the leader to rank arms based on their empirical rewards. As a result, $\mathcal{M}_p^M$ changes frequently. Combined with the delay in communication, this causes large discrepancies between $\mathcal{M}_p^j$ for different followers $j \in [M], j \neq M$, leading to frequent collisions and significant fluctuations.

We observe that the regrets of some algorithms that in our comparison increase rapidly, so we evaluated DDSE in both decentralized and centralized settings. Experimental results in Figure 10 show that performance of DDSE closely matches that in the centralized setting.

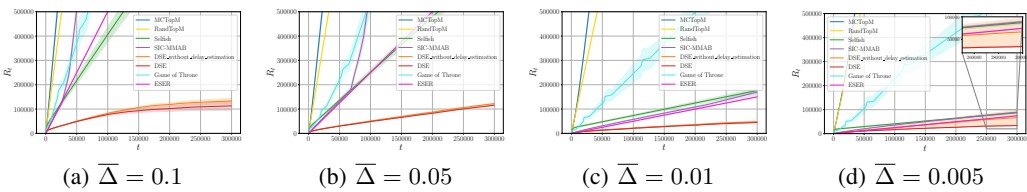

(a) $\overline{\Delta} = 0.1$  (b) $\overline{\Delta} = 0.05$  (c) $\overline{\Delta} = 0.01$  (d) $\overline{\Delta} = 0.005$

Figure 9: Comparison between different algorithms on $\overline{\Delta}$

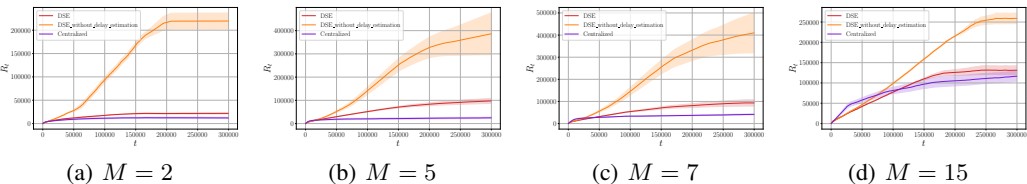

(a) $M = 2$  (b) $M = 5$  (c) $M = 7$  (d) $M = 15$

Figure 10: Comparison on $M$ with centralized algorithm

## C ALGORITHMIC DETAILS

More details about algorithms are provided in this section.

### C.1 DETAILS OF DDSE

Algorithm 4 outlines DDSE from the perspective of followers. During the exploration phase, followers select arms within $\mathcal{M}_{p-q_j}^j$ in a round-robin way and communicate with the leader every $KM \log(T)$ rounds. At the end of each communication phase, follower $j$ determines $q_j$ based on her estimation of $\hat{\mu}_d^j$ and $(\sigma_d^2)^j$, allowing her to use $\mathcal{M}_{p-q_j}^j$ in the next exploration phase. If a follower receives an ending signal from the communication phase and $q_j = 0$, she begins exploitation by continuously selecting $\mathcal{M}_{p_{\max}}^j(j)$ until $T$.

---

**Algorithm 4** DDSE (Follower)

    **Input:** $j$ (ID of each player), $K$, $M$;
1: Initialize $\mathcal{M}_0^j$ randomly, $e_j$ (ending signal), $p = 0$, $q_j = 0$, $\hat{\mu}_d^j = 0$ and $\hat{\sigma}_d^j = 0$;
2: **while** $t \leq T$ **do**
3:     Explore arms in $\mathcal{M}_{p-q_j}$;
4:     Update $\hat{\mu}_d^j$, $(\hat{\sigma}_d^2)^j$, $\mathcal{M}_{p'}$ for all $p' \leq p$;
5:     **if** $t \mod KM\lceil \log(T) \rceil = 0$ **then**               ▷ communication phase
6:         $p \leftarrow \frac{t}{KM\lceil K\log(T) \rceil}$;
7:         Communication$(j, \mathcal{M}_{p-q_j}^j)$;
8:         Find $q_j$ s.t. (1);         ▷ coordinate to the same set of best empirical arms
9:     **end if**
10:     **if** $e_j = 1$ && $q_j = 0$ **then**
11:         Select $\mathcal{M}_{p_{\max}}^j(j)$ until $T$.               ▷ exploitation phase
12:     **end if**
13: **end while**

---

If the leader chooses not to update $\mathcal{M}_{p-q_M}^M$ during the $p$-th communication phase, she conducts a virtual communication where no collision signals are sent to the followers. Algorithm 5 describes the virtual communication process, which is also divided into three parts. However, unlike real communication, the leader selects arms from $\mathcal{M}_{p-q_M}^M$ with followers in a round-robin fashion and pulls from $\mathcal{M}_{p-q_M}^M$ for $M$ rounds in the third part. Consequently, even though she has no new information to share with the followers, she remains synchronized with them.

---

**Algorithm 5** VirtualCommunication

**Input:** $\mathcal{M}_{p-q_M}^M$;
**Part 1: Remove Virtual Arm**
1: **for** $M$ time steps **do**
2:     $m_i \leftarrow [(t+M) \bmod M]$;
3:     Select $\mathcal{M}_{p-q_M}^M(m_i)$;                                     $\triangleright$ do not sent collisions
4: **end for**
   **Part 2: Add Virtual Arm**
5: $\pi \leftarrow M$;
6: **for** $K$ time steps **do**
7:     $m_i \leftarrow [(t+M) \bmod K]$;
8:     Select $m_i$;
9: **end for**
   **Part 3: Notify End**
10: **for** $M$ time step **do**
11:     Select $\mathcal{M}_{p-q_M}^M(M)$.
12: **end for**

---

## C.2 DDSE WITHOUD DELAY ESTIMATION

In this section, we provide a detailed description of DDSE_withoud_delay_estimation, which is a simplified version of Algorithm 1. By comparing this simplified version with the full algorithm, we demonstrate the superiority of the complete version.

The key difference between DDSE and this simplified algorithm is that, in Algorithm 6, the leader does not wait for the followers to select the same set of best empirical arms. Instead, she continues exploring in the latest $\mathcal{M}_p^M$ and updates it at any time. After communication, she immediately uses the new result, regardless of whether the followers have received the update.

---

**Algorithm 6** DDSE_withoud_delay_estimation (Leader)

**Input:** $K$, $M$;
1: Initialize $\mathcal{M}_0^M$ randomly, $\mathcal{K} = [K]$ $e_M = 0$ (ending signal), $p = 0$, $S_k(t) = 0$ and $\hat{\mu}_k(t) = 0$;
2: **while** $t \leq T$ **do**
3:     Explore in $\mathcal{M}_p^M$ and $\mathcal{K}/\mathcal{M}_p^M$;                          $\triangleright$ directly use the latest result
4:     Update $S_k(t), \hat{\mu}_k(t)$;
5:     Remove from $\mathcal{K}$ all arms $k$ s.t. $|\{i \in \mathcal{K} | LCB_t(i) \geq UCB_t(k)\}| \geq M$;
6:     **if** $t \bmod (KM\lceil \log(T) \rceil) = 0$ **then**
7:         $p \leftarrow \frac{t}{KM\lceil \log(T) \rceil}$;
8:         update $\mathcal{M}_p^M$;
9:         **if** $\mathcal{M}_p^M \neq \mathcal{M}_{p-1}^M$ **then**                   $\triangleright$ communication phase
10:             Communication$(a_p^-, a_p^+, i_{a_p^-}, e_M, \mathcal{M}_p^M)$;
11:         **else** VirtualCommunication$(\mathcal{M}_p^M)$;
12:         **end if**
13:     **end if**
14:     **if** $|\mathcal{K}| = M$ && $e_M = M$ **then**                       $\triangleright$ exploitation phase
15:         Select $\mathcal{M}_{p_{\max}}^M(M)$ until $T$.
16:     **end if**
17: **end while**

---

Algorithm 7 depicts the procedure from the view of followers. In the beginning, they select arms from $\mathcal{M}_0^j$ which is randomly initialized. After at least one communication phase has passed, followers check whether they have received both $a_{p'}^+$ and $i_{a_{p'}^-}$. Since $\mathcal{M}_p^j$ is maintained in a specific order, the followers place $a_{p'}^+$ at the position of $a_{p'}^-$; otherwise, the order is disrupted and followers fail to receive the update $\mathcal{M}_p^j$ by using $\mathcal{M}_{p-q_j}^j$. If both of $a_{p'}^+$ and $i_{a_{p'}^-}$ have been received, follower $j$ updates $\mathcal{M}_{p'}^j$ and uses it in the following round.

---

**Algorithm 7** DDSE_withoud_delay_estimation (Follower)

---

**Input:** $j$ (ID of each player), $K$, $M$;

1: Initialize $\mathcal{M}_0^j$ randomly, $p = 0$ and $e_j = 0$ (ending signal);
2: **while** $t \leq T$ **do**
3:    Find the latest received $\mathcal{M}_{p'}^j$;                   ▷ update the set of best empirical arms
4:    Explore in $\mathcal{M}_{p'}^j$;
5:    **if** $t \bmod (KM\lceil \log(T) \rceil) = 0$ **then**
6:        $p \leftarrow \frac{t}{KM\lceil \log(T) \rceil}$;
7:        Communication$(j, \mathcal{M}_{p'}^j)$;                   ▷ communication phase
8:    **end if**
9:    **if** $e_j = 1$ && $\mathcal{M}_{p_{\max}}^j$ has been received **then**
10:       Select $\mathcal{M}_{p_{\max}}^j(j)$ until $T$.             ▷ exploitation phase
11:   **end if**
12: **end while**

---

# D    PROOF OF THEOREM 2

## D.1    AUXILIARY LEMMAS

In this section, we provide some technical lemmas that will be useful in the proofs. The first is the well-known Hoeffding's Inequality.

**Lemma 3** *If $X_1, X_2, ..., X_n$ are sequence of i.i.d. random variables with mean $\mu$ and for every $i, X_i \in [0, 1]$, $\hat{\mu}_n := \frac{1}{n} \sum_{i \leq n} X_i$, then for all $a > 0$,*

$$P(|\hat{\mu}_n - \mu| \geq a) \leq \exp(-2na^2)$$

In addition, we need some known results about delayed feedback. The following lemma describes the relation between the received feedback and the sent feedback before $d(\theta)$.

**Lemma 4** (Lancewicki et al., 2021) *At time $t$, for any quantile $\theta \in (0, 1]$, it holds that*

$$P\left[ n_{t+d(\theta)}(k) < \frac{\theta}{2} N_t(k) \right] \leq \exp\left( -\frac{\theta}{8} N_t(k) \right),$$

where we review that $N_t(k) = \sum_{s<t} \mathbb{1}\{\pi_s^j = k, j = M\}$ is the number of times that the leader chooses arm $k$ before $t$ and $n_t(k) = \sum_{s<t} \mathbb{1}\{\pi_s^j = k, d_s^j + s < t, j = M\}$ is the number of received feedback of the leader from arm $k$ before $t$.

The overall regret can be decomposed as $R_T = R_{expl} + R_{com}$. Define $\mathcal{M}^*$ as the set of optimal arms with $|\mathcal{M}^*| = M$. We prove Theorem 2 by first analyzing the exploration phase and then the communication phase.

## D.2    EXPLORATION

In the exploration phase, Lemma 5 ensures that the delayed feedback from the communication phase of all followers is bounded. Then Lemma 6 establishes the accuracy of the estimates for $\mathbb{E}[d]$ and $\sigma_d^2$. As a result, player $j$ can correctly determine $q_j$ and align with the same best empirical arm set, thereby preventing collisions caused by inconsistencies between the leader and the followers. Thus, the regret in exploration phase is generated from (1) selecting sub-optimal arms, (2) players not receiving any feedback initially, and (3) the leader not entering the exploitation phase immediately after identifying all sub-optimal arms. During the period after the leader identifies all sub-optimal arms but before entering the exploitation phase, the leader still needs to maintain consistency with followers by selecting arms in $\mathcal{M}_{p-q_M}$, i.e., $|\mathcal{K}| = M$, $e_M = M$ but $q_M \neq 0$ which do not satisfy Line (14) in Algorithm 1.

**Lemma 5** *The feedback in one communication phase is received by all the followers after $\mathbb{E}[d] + \sqrt{2\sigma_d^2 \log((M-1)(K+M)T)}$ rounds with probability at least $1 - \frac{1}{T}$.*

*Proof*    Let $\gamma_p^j(s)$ denote the time interval in which the feedback from time $s$ during the $p$-th communication phase is fully received by player $j$ completely. We define

$$\mathcal{A} := \left\{ p \geq 1 \mid \mathcal{M}_{p-q_j}^j \neq \mathcal{M}_{p-q_M}^M, \forall j \in [M], j \leq M - 1 \right\},$$

which is the event where the best empirical arm of any follower $j$ differs from that of the leader. Then we have

$$P(\mathcal{A}) \leq \sum_{j=1}^{M-1} P(d_s^j \geq \gamma_p^j(s), ..., d_{s-(K+2M-1)}^j \geq \gamma_p^j(s) - (K + 2M - 1)) \tag{3}$$

$$\leq \sum_{j=1}^{M-1} \sum_{t=0}^{K+2M-1} P(d_{s-t}^j \geq \gamma_p^j(s) - t).$$

Due to the sub-Gaussian property of the delay,

$$P(d_s^j \geq \gamma_p^j(s)) = P(d_s^j - \mathbb{E}[d] \geq \gamma_p^j(s) - \mathbb{E}[d]) \tag{4}$$

$$\leq \exp\left( -\frac{(\gamma_p^j(s) - \mathbb{E}[d])^2}{2\sigma_d^2} \right).$$

Plug (4) into (3) and we obtain

$$P(\mathcal{A}) \leq \sum_{j=1}^{M-1} \sum_{t=0}^{K+2M-1} \exp\left(-\frac{(\gamma_p^j(s-t) - \mathbb{E}[d] - t)^2}{2\sigma_d^2}\right)$$

$$\overset{(a)}{\leq} \sum_{j=1}^{M-1} (K + 2M) \exp\left(-\frac{(\gamma_p^j(s) - \mathbb{E}[d])^2}{2\sigma_d^2}\right)$$

$$\leq (M - 1)(K + 2M) \exp\left(-\frac{(\gamma_p^j(s) - \mathbb{E}[d])^2}{2\sigma_d^2}\right),$$

where the inequality (a) is because (4) increase as $\gamma_p^j(s)$ decreases. We set the probability to $\frac{1}{T}$ and it holds that $\gamma_p^j(s) \geq \mathbb{E}[d] + \sqrt{2\sigma_d^2 \log((M-1)(K+2M)T)}$. Hence, we have proved the lemma.

$\square$

Define

$$\hat{\mu}_{d_t^j}^j := \frac{\sum_{s \leq t} \left(d_s^j \mathbb{1}\{s + d_s^j \leq t\}\right)}{\sum_{s \leq t} \mathbb{1}\{s + d_s^j \leq t\}}, \quad (\hat{\sigma}_{d_t^j}^2)^j := \frac{\sum_{s \leq t} \left((d_s^j - \hat{\mu}_{d_t^j}^j)\mathbb{1}\{s + d_s^j \leq t\}\right)^2}{\sum_{s \leq t} \mathbb{1}\{s + d_s^j \leq t\}}$$

as the estimation of $\mathbb{E}[d]$ and $\sigma_d^2$ of player $j$ by the end of time $t$. Note that $\sigma_d^2$ is the sub-Gaussian parameter, and we estimate it using the sample variance $(\hat{\sigma}_{d_t^j}^2)^j$.

**Lemma 6** *For any given $K$, $M$, $j \in [M]$ and positive integer $n$,*

$$P\left( |\hat{\mu}_{d_t^j}^j - \mathbb{E}[d]| \geq \frac{KM \log(T)}{2} \right) \leq 2 \left(\frac{1}{T}\right)^{\frac{nK^2 M^2 \log(T)}{8\sigma_d^2}},$$

$$P\left( |(\hat{\sigma}_{d_t^j}^j)^2 - \sigma_d^2| \geq \frac{K^2 M^2 \log(T)}{40} \right) \leq 2 \left(\frac{1}{T}\right)^{\frac{nK^2 M^2}{320\sqrt{2}\sigma_d^2}}.$$

*Proof*    After the first communication phase, players begin to select a previous best empirical arm set based on $\hat{\mu}_{d_t^j}^j$ and $\hat{\sigma}_{d_t^j}^j$. Next, we consider how large the error in $\hat{\mu}_{d_t^j}^j$ and $\hat{\sigma}_{d_t^j}^j$ leads to different sets of best empirical arms. Since the goal of player $j$ is to find a backward counting number $q_j$ s.t.

$$t > \hat{\mu}_{d_t^j}^j + \sqrt{2(\hat{\sigma}_{d_t^j}^j)^2 \log((M-1)(K+2M)T)} + (p - q_j)KM \log(T), \tag{5}$$

only when the error of (5) reach $KM \log(T)$, it will lead that player $j$ chooses a wrong $\mathcal{M}_{p-q_j}^j$. Define the error of $\mathbb{E}[d]$ as $\epsilon_\mu$ and the error of $\sigma_d^2$ as $\epsilon_\sigma$.

We know that $d_t^j \sim \text{sub-G}(\sigma_d^2)$, so $\hat{\mu}_{d_t^j}^j$ is also a sub-Gaussian variable with parameter $\sigma_d^2/n_t$, where $n_t := \sum_{k \in [K]} \sum_{s \le t} \mathbb{1}\{d_s^j + s < t, \pi_s^j = k\}$ is the total number that feedbacks are received by the end of time $t$. Thus, $(d_t^j - \hat{\mu}_{d_t^j}^j)$ is a sub-Gaussian variable with parameter $\sigma_d^2(1+1/n_t)$. According to Lemma 2.7.5 in Vershynin (2018), the product of sub-Gaussian variables is sub-Exponential, which implies that $(\hat{\sigma}_{d_t^j}^2)^j$ is sub-Exponential. The tail-bound for the sub-Exponential variable $(\hat{\sigma}_{d_t^j}^2)^j$ with parameter $(v^2, \alpha)$ is

$$P(|(\hat{\sigma}_{d_t^j}^2)^j - \sigma_d^2| \ge \epsilon_\sigma) \le 2 \exp\left(-\frac{1}{2} \min\left\{\frac{n_t \epsilon_\sigma^2}{v^2}, \frac{n_t \epsilon_\sigma}{\alpha}\right\}\right). \tag{6}$$

We consider three situations. The first is $\hat{\mu}_{d_t^j}^j$ is very close to $\mathbb{E}[d]$ but the error of $(\hat{\sigma}_{d_t^j}^2)^j$ is quite large. By (6), large $\epsilon_\sigma$ leads to the probability of $|(\hat{\sigma}_{d_t^j}^2)^j - \sigma_d^2| \ge \epsilon_\sigma$ is small. This means that the probability of significant deviation on $(\hat{\sigma}_{d_t^j}^2)^j$ is extremely low, leading to almost correct $\hat{\mu}_{d_t^j}^j$ and $(\hat{\sigma}_{d_t}^2)^j$ with high probability. Another situation is that $(\hat{\sigma}_{d_t^j}^2)^j$ is very close to $\sigma_d^2$ but the error of $\hat{\mu}_{d_t^j}^j$ is large. The analysis is similar to the first situation because $\hat{\mu}_{d_t^j}^j$ is a sub-Gaussian variable. Therefore, when $\hat{\mu}_{d_t^j}^j$ and $(\hat{\sigma}_{d_t^j}^2)^j$ have errors of

$$\epsilon_\mu \ge \frac{KM \log(T)}{2},$$
$$\epsilon_\sigma(2 \log((M-1)(K+2M)T)) \ge \left(\frac{KM \log(T)}{2}\right)^2, \tag{7}$$

incorrect $q_j$ will occur. Since $M \le K$, we have

$$\epsilon_\sigma \ge \frac{K^2 M^2 (\log(T))^2}{8(\log(3KMT))}$$
$$= \frac{K^2 M^2 (\log(T))^2}{8(\log(M) + 3\log(K) + \log(T))}$$
$$\ge \frac{K^2 M^2 \log(T)}{40}.$$

The gap between estimated mean $\hat{\mu}_d^j$ and expectation of delay $\mathbb{E}[d]$ is bounded as

$$P(|\hat{\mu}_{d_t}^j - \mathbb{E}[d]| \ge \epsilon_\mu) \le 2 \exp(-\frac{n_t \epsilon_\mu^2}{2\sigma_d^2}).$$

Plug (7) and it holds that

$$P\left(|\hat{\mu}_{d_t}^j - \mathbb{E}[d]| \ge \frac{KM \log(T)}{2}\right) \le 2 \exp(-\frac{n_t K^2 M^2 (\log(T))^2}{8\sigma^2})$$
$$\le 2\left(\frac{1}{T}\right)^{\frac{n K^2 M^2 \log(T)}{8\sigma^2}}.$$

From Honorio & Jaakkola (2014) we have $v^2 = 4\sqrt{2}\sigma_d^2(1+1/n_t)$ and $\alpha = 4\sigma_d^2(1+1/n_t)$. Thus, it holds that

$$P\left(|\hat{\sigma}_{d_t}^j - \sigma_d| \ge \frac{K^2 M^2 \log(T)}{40}\right) \le 2 \exp\left(-\frac{1}{2} \min\left\{\frac{n_t \epsilon_\sigma^2}{4\sqrt{2}\sigma_d^2(1+1/n_t)}, \frac{n_t \epsilon_\sigma}{4\sigma_d^2(1+1/n_t)}\right\}\right)$$
$$\le 2 \exp\left(-\frac{n_t^2 K^2 M^2 \log(T)}{160\sqrt{2}\sigma_d^2(n_t+1)}\right)$$
$$\le 2\left(\frac{1}{T}\right)^{\frac{n K^2 M^2}{320\sqrt{2}\sigma_d^2}},$$

where the last inequality comes from $n_t \geq 1$.

$\square$

**Lemma 7 (Restatement of Lemma 1)** *In decentralized setting, for delay distribution under Assumption 1, given any $K, M, \mu$ and a quantile $\theta \in (0, 1]$, the regret of the exploration phase in DDSE is bounded by*

$$R_{expl} \leq \sum_{k>M} \frac{323 \log(T)}{\theta \Delta_k} + \sum_{k>M} \frac{M\Delta_k}{K-M} \left( 2\mathbb{E}[d] + \sigma_d \sqrt{\log((M-1)(K+2M))} \right)$$

$$+ 3\sigma_d \sqrt{2 \log \left( \frac{1}{1-\theta} \right)} + 3\mathbb{E}[d] + C_2,$$

*where $C_2 = \frac{4Me^{-\delta^2/2}}{\delta^2}$*

*Proof*    We have proved that in the exploration phase, players coordinate to the same set of best empirical arms and collisions do not happen with high probability. Thus, when players are in the exploration phase, the regret is only due to the selection of sub-optimal arms and delays. Define the following events:

$$\mathcal{B} := \left\{ t \geq 1 \mid \mathcal{M}_p^j \neq \mathcal{M}^*, \forall j \in [M], p = \left\lceil \frac{t}{KM \log(T)} \right\rceil, 1 \leq p \leq p_{\max} \right\},$$

$$\mathcal{C} := \left\{ t \geq 1 \mid \exists k \in [K] \text{ s.t. } |\hat{\mu}_t(k) - \mu_t(k)| \geq \sqrt{\frac{2 \log(T)}{n_t(k)}} \right\},$$

$$\mathcal{D} := \left\{ t \geq 1 \mid \exists k \in [K] \text{ s.t. } N_t(k) \geq \frac{32 \log(T)}{\theta}, n_{t+d(\theta)}(k) \leq \frac{\theta}{2} N_t(k) \right\},$$

$$\mathcal{E} := \left\{ t \geq 1 \mid \mathcal{M}_{p_{\max}}^M = \mathcal{M}^*, \mathcal{M}_{p_{\max}}^M \neq \mathcal{M}_{p_{\max}}^j, \exists j \leq M-1 \right\}.$$

Here $\mathcal{B}$ means that the best empirical arm set of players is different from $\mathcal{M}^*$ by the time step $t$. $\mathcal{C}$ represents the occurrence of a bad event where successive elimination leads to an incorrect result. $\mathcal{D}$ means that the received feedback after after $d(\theta)$ is insufficient. $\mathcal{E}$ indicates that the leader has already identified $\mathcal{M}^*$ but at least one follower has not yet received feedback from the final communication phase by the time step $t$.

Recall that $\delta = \min_{1 \leq k \leq K-1}(\mu_{(k)} - \mu_{(k+1)})$ is the minimum gap between the rewards of arms. When the error between $\hat{\mu}_k$ and $\mu_k$ less than $\delta/2$, the leader can distinguish each arm by their empirical rewards. Denote $T_{expl}$ as the total time of the exploration phase. Define player $j$ does not receive feedback before $d_{\tilde{t}}^j$. By Lemma 3, we have

$$\mathbb{E}[|\mathcal{B}|] \leq \mathbb{E} \left[ \sum_{t=\tilde{t}}^{T_{expl}} 2 \exp(-2t(\frac{\delta}{2})^2) + d_{\tilde{t}}^j \right]$$

$$\leq \int_{\tilde{t}}^{+\infty} 2 \exp(-\frac{\delta^2 t}{2}) \, dt + \mathbb{E}[d_{\tilde{t}}^j] \tag{8}$$

$$\leq \frac{4e^{-\tilde{t}\delta^2/2}}{\delta^2} + \mathbb{E}[d]$$

$$\leq \frac{4e^{-\delta^2/2}}{\delta^2} + \mathbb{E}[d].$$

We can bound $\mathbb{E}[|\mathcal{C}|] \leq 2T^{-1}$ by directly using Lemma 3. From Lemma 4 and a union bound, we also have $\mathbb{E}[|\mathcal{D}|] \leq T^{-1}$. Next, we give some definitions that are similar to Lancewicki et al. (2021). Denote $t_\ell$ as the time that the leader pulled all the active arms after $32 \log(T)/\theta \epsilon_\ell^2$ times where $\epsilon_\ell = 2^{-\ell}$. Define $0 \leq \kappa_\ell \leq K$ such that $t_\ell + d(\theta) + \kappa_\ell$ is an elimination step. Let $\mathcal{S}_\ell$ be the set of sub-optimal arms, that were not eliminated by time $t_\ell + d(\theta) + \kappa_\ell$, but were eliminated by

time $t_\ell + d(\theta) + \kappa_{\ell+1}$. Then the regret of eliminating sub-optimal arms is bounded as

$$R_{elm} \leq \sum_{t=1}^{t_0+d(\theta)+\kappa_0} \sum_{k>M} \mathbb{1}\{\pi_t^M = k\}\Delta_k + \sum_{\ell=0}^{\infty} \left( 3\left(d(\theta) + K\right)\epsilon_{\ell+1} + \sum_{k \in S_\ell} N_{t_\ell+d(\theta)+\kappa_\ell}(k)\Delta_k \right)$$

$$\leq \frac{32\log(T)}{\theta}(K - M) + 3\left(d(\theta) + K\right) + \sum_{k>M} \frac{288\log(T)}{\theta\Delta_k}$$

$$\leq \sum_{k>M} \frac{323\log(T)}{\theta\Delta_k} + 3d(\theta),$$

(9)

which is a direct consequence of Theorem 2 in Lancewicki et al. (2021). Recall that $d(\theta) = \min\{\gamma \in \mathbb{N} \mid P(d \leq \gamma) \geq \theta\}$. By Assumption 1,

$$P(d_t^j \leq d(\theta)) = 1 - P\left(d_t^j - \mathbb{E}[d] \geq d(\theta) - \mathbb{E}[d]\right)$$

$$\geq 1 - \exp\left(-\frac{(d(\theta) - \mathbb{E}[d])^2}{2\sigma^2}\right).$$

Thus, we have $\theta \geq 1 - \exp(-(d(\theta) - \mathbb{E}[d])^2/2\sigma^2)$ and

$$d(\theta) \leq \sqrt{2\sigma_d^2 \log\left(\frac{1}{1-\theta}\right)} + \mathbb{E}[d].$$

(10)

After the leader identifies $\mathcal{M}^*$, she still needs to coordinate to use $\mathcal{M}_{p-q_M}^M$ and wait for followers who have not received the last feedback. Define $\mathcal{T}_e = \{t_e, ..., t_e + K + 2M\}$ as the final communication phase. Feedbacks from $\mathcal{T}_e$ will be received after $\max_{t \in T_e, j \leq M-1} d_t^j$. Then $\mathbb{E}[|\mathcal{E}|]$ is bounded as

$$\mathbb{E}[|\mathcal{E}|] \leq \mathbb{E}\left[\max_{t \in \mathcal{T}_e, j \leq M-1} d_t^j\right].$$

By Jensen's inequality, we have

$$\exp\left(\lambda\mathbb{E}[\max_{t \in \mathcal{T}_e, j \leq M-1} d_t^j]\right) \leq \mathbb{E}[\exp(\lambda \max_{t \in \mathcal{T}_e, j \leq M-1} d_t^j)]$$

$$= \mathbb{E}[\max_{t \in \mathcal{T}_e, j \leq M-1} \exp(\lambda d_t^j)]$$

$$\leq \mathbb{E}[\sum_{t=1}^{K+2M}\sum_{j=1}^{M-1} \exp(\lambda d_t^j)],$$

(11)

Since $d_t^j - \mathbb{E}[d]$ is a sub-Gaussian variable with zero expectation, $\mathbb{E}[\exp(\lambda(d_t^j - \mathbb{E}[d]))] \leq \exp(\lambda^2\sigma^2/2)$. Therefore, we have

$$\mathbb{E}[\exp(\lambda d_t^j)] \leq \exp\left(\frac{\lambda^2\sigma^2}{2} + \lambda\mathbb{E}(d)\right).$$

Plug it into (11) and we get

$$\exp\left(\lambda\mathbb{E}[\max_{t \in \mathcal{T}_e, j \leq M-1} d_t^j]\right) \leq (M-1)(K+2M)\exp\left(\frac{\lambda^2\sigma^2}{2} + \lambda\mathbb{E}(d)\right).$$

When $\lambda^* = \sqrt{2\log((M-1)(K+2M))}/\sigma_d$,

$$\mathbb{E}[|\mathcal{E}|] \leq \mathbb{E}\left[\max_{t \in \mathcal{T}_e, j \leq M-1} d_t^j\right]$$

$$\leq \sigma_d\sqrt{2\log((M-1)(K+2M))} + \mathbb{E}[d].$$

(12)

Finally, the regret in the exploration phase can be bound as

$$R_{expl} \leq R_{elm} + M\mathbb{E}[\mathcal{B} \cup \mathcal{E}]\frac{\sum_{k>M}\Delta_k}{K-M}.$$

Plug (8), (9), (10), (12) and we have

$$R_{expl} \leq \sum_{k>M} \frac{323\log(T)}{\theta\Delta_k} + 3\sigma_d\sqrt{2\log\left(\frac{1}{1-\theta}\right)} + 3\mathbb{E}[d] + \frac{4Me^{-\delta^2/2}}{\delta^2}$$
$$+ \sum_{k>M} \frac{M\Delta_k}{K-M}\left(2\mathbb{E}[d] + \sigma_d\sqrt{\log((M-1)(K+2M))}\right).$$

$\square$

### D.3 COMMUNICATION

We have already known that the length of each communication phase is $K + 2M$. Note that player enter a communication phase evert $KM\log(T)$ rounds. The next step is to bound the times that the leader need to receive feedback and eliminate all sub-optimal arms.

**Lemma 8 (Restatement of Lemma 2)** *In decentralized setting, for delay distribution under Assumption 1, given any $K, M, \mu$ and a $\theta \in (0,1]$, the regret in the communication phase is bounded by*

$$R_{com} \leq \sum_{k>M} \frac{195}{\theta\Delta_k^2} + 3\sigma_d\left(\sqrt{6} + \sqrt{2\log(\frac{1}{1-\theta})}\right) + 6\mathbb{E}[d].$$

*Proof*    Denote the time that the leader need to receive feedback and eliminate all sub-optimal arms by $T_{expl}$. After $T_{expl}$, the leader has waited for the followers for $\mathbb{E}[\max_{t\in T_e, j\in[M]} d_t^j]$ rounds after eliminating all sub-optimal arms. Consider a sub-optimal arm $k$ which has not been eliminated at time $\tau_k$, but remains active until the next exploration phase ends. From the arm elimination condition, the gap between the arm $k_M$ with $\mu_{(M)}$ which is the $M$-th reward mean and $k$ is bounded by

$$\Delta_k \leq 2\left[\sqrt{\frac{2\log(T)}{n_{\tau_k}(k)}} + \sqrt{\frac{2\log(T)}{n_{\tau_k}(k_M)}}\right]$$
$$\leq 2\left[\sqrt{\frac{2\log(T)}{n_{\tau_k}(k)}} + \sqrt{\frac{2\log(T)}{n_{\tau_k}(k)}}\right] \tag{13}$$
$$\leq 4\sqrt{\frac{2\log(T)}{n_{\tau_k}(k)}}.$$

Since $\mathbb{E}[|\mathcal{D}|]$ is bounded by $T^{-1}$, $n_{\tau_k}(k) \geq \theta N_{\tau_k-d(\theta)}(k)/2$ and $N_{\tau_k-d(\theta)} \leq 64\log(T)/\theta\Delta_k^2$. Thus, $T_{expl}$ is bounded as

$$T_{expl} \leq \sum_{k>M} N_{\tau_k}(k) + \mathbb{E}[\max_{t\in T_e, j\in[M]} d_t^j]$$
$$\leq \sum_{k>M}\left(1 + N_{\tau_k-d(\theta)}(k) + \underbrace{N_{\tau_k}(k) - N_{\tau_k-d(\theta)}(k)}_{d(\theta)}\right) + \mathbb{E}[\max_{t\in T_e, j\in[M]} d_t^j].$$

Define $f(k)$ as the number of active arms at $\tau_k$. By $K > M$, we have

$$\sum_{k>M} \frac{1}{f(k)} = \sum_{j=0}^{M} \frac{1}{K-j}$$
$$\leq \log(K) - \log(K-M)$$
$$\leq \log(K)$$

Note that the leader makes selection over all active arms, so it holds that

$$
\begin{aligned}
T_{expl} &\leq \sum_{k>M} \frac{65\log(T)}{\theta\Delta_k^2} + \mathbb{E}[\sum_{k>M} \frac{d(\theta)}{f(k)}] + \mathbb{E}[\max_{t\in T_e, j\in[M]} d_t^j] \\
&\leq \sum_{k>M} \frac{65\log(T)}{\theta\Delta_k^2} + (1+\log(K))\,\mathbb{E}[d] + \sigma_d(\log(K)\sqrt{2\log(\frac{1}{1-\theta})} \\
&\quad + \sqrt{2\log((M-1)(K+2M))}).
\end{aligned}
\tag{14}
$$

The leader communicates with followers every $KM\log(T)$ times, so the total communication time $T_{com}$ is $T_{expl}/KM\log(T)$. Then the regret of communication can be bounded as $R_{com} \leq (K+2M)MT_{com}$, where we take a union bound of $M$ players. Since $M \leq K$, we have $R_{com} \leq 3KMT_{com}$. Plugging in (10) and (14), we have the following result:

$$
\begin{aligned}
R_{com} &\leq 3KM \frac{T_{expl}}{KM\log(T)} \\
&\leq \sum_{k>M} \frac{195}{\theta\Delta_k^2} + \frac{3log(K)}{\log(T)}\left(2\mathbb{E}[d] + \sigma_d\sqrt{2\log(\frac{1}{1-\theta})}\right) + \frac{3\sigma_d\sqrt{6\log(K)}}{\log(T)} \\
&\leq \sum_{k>M} \frac{195}{\theta\Delta_k^2} + 3\sigma_d\left(\sqrt{6} + \sqrt{2\log(\frac{1}{1-\theta})}\right) + 6\mathbb{E}[d].
\end{aligned}
\tag{15}
$$

$\square$

# E    PROOF OF COROLLARY 1

*Proof*    In centralized setting, players can freely communicate with each other so we do not need the communication phase and (15) vanishes. Due to the centralized setting, followers constantly know the latest exploration result of the leader, leading to $\mathcal{M}_p^j = \mathcal{M}_p^\ell, \forall j \in [M], \ell \in [M], p \leq p_{\max}$. Thus, there is no need for the leader to wait for followers to receive the final feedback in $\mathcal{T}_e$ and the regret caused by $\mathbb{E}[|\mathcal{E}|]$ in (12) also disappears. The regret of DDSE in centralized setting is

$$
R_T \leq R_{elm} + M\mathbb{E}[|\mathcal{B}|]\frac{\sum_{k>M}\Delta_k}{K-M}.
$$

Plug (8), (9) and we obtain the result.

$\square$

# F    PROOF OF THEOREM 3

Denote $R'_{expl}$ as the regret of DDSE_without_delay_estimation in exploration phase and $R'_{com}$ as the regret of DDSE_without_delay_estimation in communication phase. We decompose the total regret as $R_T = R'_{expl} + R'_{com}$.

**Lemma 9** *In decentralized setting, for delay distribution under Assumption 1, given any $K, M, \mu$ and a quantile $\theta \in (0, 1]$, the regret of DDSE_without_delay_estimation in exploration phase is bounded by*

$$
\begin{aligned}
R'_{expl} &\leq \sum_{k>M} \frac{323\log(T)}{\theta\Delta_k} + (3 + \frac{M\sum_{k>M}\Delta_k}{K-M})\mathbb{E}[d] + 3\sigma_d\sqrt{2\log(\frac{1}{1-\theta})} \\
&\quad + \exp\left(\frac{\mathbb{E}[d]}{KM} + \frac{\sigma_d^2}{2K^2M^2}\right) + C_2,
\end{aligned}
$$

*where $C_2 = \frac{4Me^{-\delta^2/2}}{\delta^2}$.*

*Proof* We define player $j$ selects a certain arm at $s_p$ in the $p$-th communication phase. Then after a period of time $d_{s_p}$, she receives $< r_{s_p}^j, \eta_{s_p}^j, s >$ at $t_{s_p}$. Then we define

$$\mathcal{F} := \{t_{s_{p-1}} \mid t_{s_{p-1}} > t_{s_p}, \forall p \in \mathcal{T}_{com}\},$$

which indicates that at least one feedback from the $(p-1)$-th communication phase has not been received by the time the feedback from the $p$-th communication phase is received. Then after certain time $\gamma_p$, the probability that player $j$ receive the feedback from $s_p$ is $P(d_{s_p} \leq \gamma_p)$. The probability that certain feedback in phase $p-1$ has not been received is $P(d_{s_{p-1}} \geq \gamma_p + s_p - s_{p-1})$. Denote $R_{\mathcal{F}}$ as the regret if $\mathcal{F}$ happens. Then we have

$$
\begin{aligned}
R_{\mathcal{F}} &= \mathbb{E}[P(\mathcal{F})T] \\
&= \mathbb{E}[P(d_{s_p} \leq \gamma_p)P(d_{s_{p-1}} \geq \gamma_p + s_p - s_{p-1})]T \\
&\leq \mathbb{E}[P(d_{s_{p-1}} \geq \gamma_p + s_p - s_{p-1})]T.
\end{aligned}
\tag{16}
$$

By Assumption 1, it holds that

$$
\begin{aligned}
P(d_{s_{p-1}} \geq \gamma_p + s_p - s_{p-1}) &= P(d_{s_{p-1}} - \mathbb{E}[d] \geq \gamma_p + s_p - s_{p-1} - \mathbb{E}[d]) \\
&\leq \exp\left(-\frac{(\gamma_p + s_p - s_{p-1} - \mathbb{E}[d])^2}{2\sigma_d^2}\right).
\end{aligned}
\tag{17}
$$

Plug (17) into (16) and $R_{\mathcal{F}}$ can be bounded as

$$
\begin{aligned}
R_{\mathcal{F}} &\leq \mathbb{E}\left[\exp\left(-\frac{(\gamma_p + s_p - s_{p-1} - \mathbb{E}[d])^2}{2\sigma_d^2}\right)\right]T \\
&\overset{(a)}{\leq} \exp\left(-\mathbb{E}\left[\frac{(\gamma_p + s_p - s_{p-1} - \mathbb{E}[d])^2}{2\sigma_d^2}\right]\right)T \\
&\overset{(b)}{\leq} \exp\left(-\frac{(\mathbb{E}[\gamma_p + s_p - s_{p-1} - \mathbb{E}[d]])^2}{2\sigma_d^2}\right)T \\
&\leq \exp\left(-\frac{(\mathbb{E}[s_p - s_{p-1} - \mathbb{E}[d]])^2}{2\sigma_d^2}\right)T.
\end{aligned}
$$

Here, inequalities (a) and (b) follow from Jensen's inequality. Since players enter communication phase every $KM\log(T)$ rounds, we have $\mathbb{E}[s_p - s_{p-1}] = KM\log(T)$. Note that $\mathbb{E}[d]$ is a constant about delay. Thus, we have

$$
\begin{aligned}
R_{\mathcal{F}} &\leq \exp\left(-\frac{(\mathbb{E}[s_p - s_{p-1}] - \mathbb{E}[d])^2}{2\sigma_d^2}\right)T \\
&= \exp\left(-\frac{(KM\log(T) - \mathbb{E}[d])^2}{2\sigma_d^2}\right)T.
\end{aligned}
\tag{18}
$$

We perform a change of variables by denoting $x = \log(T)$ so that $T = e^x$. Then (18) is equals to $\exp(x - \frac{(KMx - \mathbb{E}[d])^2}{2\sigma^2})$, which achieves its maximum when $x^* = \frac{\mathbb{E}[d]}{KM} + \frac{\sigma_d^2}{K^2M^2}$. Therefore, the regret that bad event $\mathcal{E}$ happens is bounded as

$$R_{\mathcal{F}} \leq \exp\left(\frac{\mathbb{E}[d]}{KM} + \frac{\sigma_d^2}{2K^2M^2}\right).
\tag{19}
$$

When $\mathcal{F}$ does not occur, players can communicate successfully and remain synchronized while exploring the set of best arms. Since players always use the latest result of communication that they have received, the leader does not wait for followers after she identifies $\mathcal{M}^*$ and the regret caused by $\mathcal{E}$ vanishes. Then the regret of DDSE_without_delay_estimation in exploration phase is bounded by

$$R'_{expl} \leq R_{elm} + M\mathbb{E}[|\mathcal{B}|]\frac{\sum_{k>M}\Delta_k}{K-M} + R_{\mathcal{F}}.$$

Apply (8), (9), (19) and we have completed the proof.

$\square$

**Lemma 10** *In decentralized setting, for delay distribution under Assumption 1, given any $K, M, \mu$ and a quantile $\theta \in (0, 1]$, the regret of DDSE_without_delay_estimation in communication phase is bounded by*

$$R'_{com} \leq \sum_{k>M} \frac{195}{\theta \Delta_k^2} + 3\sigma_d \left( \sqrt{6} + \sqrt{2 \log(\frac{1}{1-\theta})} \right) + 6\mathbb{E}[d]$$

$$+ O\left( \frac{\tilde{d}_2 \tilde{d}_3}{KM} + \frac{\tilde{d}_3}{\theta K M \sum_{k>M} \Delta_k^2} \right),$$

*where $\tilde{d}_2 = \mathbb{E}[d] + \sqrt{\sigma_d^2 \log(1/(1-\theta))}$ and $\tilde{d}_3 = \mathbb{E}[d] + \sqrt{\sigma_d^2 \log(K)}$.*

*Proof*   In our algorithm, although players use their latest sets of best empirical arms, during the period after communication ends, their sets of best empirical arms still differ, which leads to collisions. Specifically, after the leader updates $\mathcal{M}_p^M$ and passes the update to followers, it takes time for them to receive the information because it is delayed. However, since the leader continues to use the latest $\mathcal{M}_p^M$, she will collide with followers before they receive the update of $\mathcal{M}_p^j$. Define

$$\mathcal{H} := \left\{ t \geq 1 \mid \mathcal{M}_p^M \neq \mathcal{M}_p^j, \exists j \leq M-1, p = \left\lceil \frac{t}{KM \log(T)} \right\rceil \right\}.$$

Since the length of every communication phase is $K + 2M$, by applying the same technique in (12), we have

$$\mathbb{E}[|\mathcal{H}|] \leq \sigma_d \sqrt{2 \log((M-1)(K+2M))} + \mathbb{E}[d].$$

Note that after each communication phase, at most one arm changes. Players select arms in a round-robin fashion, resulting in collisions every $M$ rounds, as one round-robin cycle consists of $M$ steps. Then taking a union bound over the $M$ players, we have

$$R_{col} \leq \mathbb{E}[|\mathcal{H}|] \frac{MT_{com}}{M}$$

$$\leq \mathbb{E}[|\mathcal{H}|] \frac{R_{com}}{3KM}.$$

Plug (14),

$$R_{col} \leq \left( \sigma_d \sqrt{2 \log((M-1)(K+2M))} + \mathbb{E}[d] \right) \frac{\left( \sum_{k>M} \frac{195}{\theta \Delta_k^2} + 3\sigma_d \left( \sqrt{6} + \sqrt{2 \log(\frac{1}{1-\theta})} \right) + 6\mathbb{E}[d] \right)}{3KM}$$

$$= O\left( \frac{\tilde{d}_2 \tilde{d}_3}{KM} + \frac{\tilde{d}_3}{\theta K M \sum_{k>M} \Delta_k^2} \right).$$

Finally, the regret of DDSE_without_delay_estimation in communication phase is bounded by

$$R'_{com} \leq R_{com} + R_{col}.$$

Also by applying (14), we have completed the proof.

$\square$

# G   PROOF OF THEOREM 1

Denote $ALG^{mmab}$ as the algorithm of a centralized multi-player multi-armed bandit problem and $R_T^{mmab}$ as the regret of $ALG^{mmab}$. We also denote $ALG^{delay}$ as the algorithm of a centralized MMAB with delayed feedback. Denote $R_T^{delay}$ as the regret of $ALG^{delay}$. Anantharam et al. (1987) proved that any strongly consistent algorithm satisfies

$$R_T^{mmab} \geq \sum_{k>M} \frac{(1 - o(1))(\mu_{(M)} - \mu_{(k)})}{D_{KL}(\mu_{(k)}, \mu_{(M)})} \log(T).$$

where $D_{KL}(\mu_{(k)}, \mu_{(M)})$ is the KL-divergence. Then by inverse Pinsker's inequality, we have

$$R_T^{mmab} \geq \sum_{k>M} \frac{(1-o(1))\log(T)}{(\mu_{(M)} - \mu_{(k)})}. \tag{20}$$

In centralized setting, the best empirical arm set of each player is the same, so we have $\mathcal{M}_p = \mathcal{M}_p^j = \mathcal{M}_p^l, \forall j \in [M], \ell \in [M], j \neq \ell$. Since there is no communication phase and players update $\mathcal{M}_p$ at every time, we change our notation into $\mathcal{M}_t$ which denotes the selected arm set by players at $t$. Define $X_t(k) \sim \text{Bernoulli}(\theta)$ as the delay choice from the selected arm $k$ at time $t$. We consider Algorithm 8 which is a variant of Lancewicki et al. (2021).

---

**Algorithm 8** Simulate Delay for Centralized MMAB

**Input:** $\theta, T$
1: Initialize $j$ (the ID of the player), $T_x = \lfloor T(1-\theta/4) \rfloor$, $X_t(k) = 0$, $S^j(t) = 0$
2: **for** $t \leq T_x$ **do**
3:     Player $j$ in $ALG^{delay}$ selects arm $k \in \mathcal{M}_t$
4:     Environment generates $X_t(k) \sim \text{Bernoulli}(\theta)$
5:     $S^j(t) \leftarrow S^j(t) + X_t(k)$
6:     **if** $X_t^j = 1$ **then**
7:         Player $j$ in $ALG^{mmab}$ selects arm $k$ and gets a reward $r^j(t)$
8:         Player $j$ in $ALG^{mmab}$ updates $r^j(t)$
9:     **end if**
10:     **if** $t = T_x$ && $S^j(t) \leq \frac{qT}{4}$ **then**
11:         $T_x \leftarrow T$
12:     **end if**
13: **end for**

---

Algorithm 8 is from the view of player $j$ in centralized MMAB. Since players can freely communicate with others, no collision will occur. Define

$$\mathcal{I} := \left\{ t \geq 1 \mid \sum_{t=1}^{\lfloor T(1-\theta/4) \rfloor} \sum_{j \in [M]} \mathbb{1}\{\pi_t^j = k\} X_t(k) \leq \frac{\theta MT}{4} \right\}.$$

We have the following lemma.

**Lemma 11** *For $\theta \in (0,1]$, any $M$, $T$, $P(\mathcal{I}) \leq \exp(-\frac{\theta MT}{16})$.*

*Proof* Define $\epsilon_{\mathcal{I}} := 1 - \frac{1}{4(1-\theta/4)}$. Since $X_t(k) \sim \text{Bernoulli}(\theta)$, $\mathbb{E}[\sum_{t=1}^{\lfloor T(1-\theta/4) \rfloor} \sum_{j \in [M]} \mathbb{1}\{\pi_t^j = k\} X_t(k)] = \theta MT(1-\theta/4)$. By Chernoff bound,

$$P(\mathcal{I}) \leq \exp\left(-\frac{\epsilon_{\mathcal{I}}^2}{2}(\theta MT(1-\frac{\theta}{4}))\right)$$

$$\leq \exp(-\frac{\theta MT}{16}),$$

where the last inequality is due to $1 - \theta/4 \geq 1/2$. $\qquad\square$

Note that $ALG^{mmab}$ only runs when $X_t^j = 1$. Therefore, we only account for the regret $R_{\lfloor \frac{1}{4} T\theta \rfloor}^{mmab}$, where $\lfloor \frac{1}{4} T\theta \rfloor$ denotes the total time $ALG^{mmab}$ is active, rather than referring to the time interval from 1 to $\lfloor \frac{1}{4} T\theta \rfloor$. The regret of $ALG^{mmab}$ is bounded by

$$R_{\lfloor \frac{1}{4} T\theta \rfloor}^{mmab} \leq \mathbb{E}\left[ \sum_{t < T(1-\theta/4)} \sum_{k>M} \mathbb{1}\{X_t(k) = 1\}\mathbb{1}\{k \in \mathcal{M}_t\}\Delta_k \right]$$

$$+ \mathbb{E}\left[ \mathbb{1}\{\mathcal{I}\} \sum_{t \geq T(1-\theta/4)} \sum_{k>M} \mathbb{1}\{k \in \mathcal{M}_t\}\Delta_k \right]$$

$$\leq \sum_{t < T(1-\theta/4)} \sum_{k>M} \mathbb{E}[\mathbb{1}\{X_t(k) = 1\}]\mathbb{E}[\mathbb{1}\{k \in \mathcal{M}_t\}\Delta_k] + \frac{\theta MT}{4} P(\mathcal{I}).$$

Since $X_t(k) \sim \text{Bernoulli}(\theta)$, we have $\mathbb{E}[\mathbb{1}\{X_t(k) = 1\}] = \theta$ for $\forall t \leq T, k \in [K]$ and it holds that

$$R_{\lfloor \frac{1}{4}T\theta \rfloor}^{mmab} \leq \theta \mathbb{E} \left[ \sum_{t < T(1-\theta/4)} \sum_{k > M} \mathbb{1}\{k \in \mathcal{M}_t\}\Delta_k \right] + 4$$

$$\leq \theta \mathbb{E} \left[ \sum_{t \leq T} \sum_{k > M} \mathbb{1}\{k \in \mathcal{M}_t\}\Delta_k \right] + 4$$

$$\leq \theta R_T^{delay} + 4.$$

Plugging (20), the regret of centralized MMAB with delay can be bounded as

$$R_T^{delay} \geq \sum_{k > M} \frac{(1 - o(1))\log(T)}{\theta \Delta_k} - \frac{4}{\theta}. \tag{21}$$

Consider a fixed delay distribution with expectation $\mathbb{E}[d]$ and variance $\sigma_d^2$:

$$P(d_t^j = x) = \begin{cases} \theta & x = \mathbb{E}[d] + \sigma_d\sqrt{\frac{1-\theta}{\theta}} \\ 1 - \theta & x = \mathbb{E}[d] - \sigma_d\sqrt{\frac{\theta}{1-\theta}} \end{cases}, \forall t \leq T, j \in [M].$$

The algorithm does not receive feedback for at least the first $(\mathbb{E}[d] - \sigma_d\sqrt{\theta/1-\theta})$ rounds and the probability of selecting a sub-optimal arm is $K - M/K$. Thus, the regret can also be bounded as

$$R_T^{delay} \geq \mathbb{E} \left[ \sum_{t=1}^{(\mathbb{E}[d] - \sigma_d\sqrt{\theta/1-\theta})} \sum_{k > M} \mathbb{1}\{k \in \mathcal{M}_t\}\Delta_k \right]$$

$$\geq \left( \mathbb{E}[d] - \sigma_d\sqrt{\frac{\theta}{1-\theta}} \right) \frac{K - M}{K} M \frac{\sum_{k > M}\Delta_k}{K - M}$$

$$= \left( \mathbb{E}[d] - \sigma_d\sqrt{\frac{\theta}{1-\theta}} \right) \frac{M}{K} \sum_{k > M} \Delta_k.$$

Combining this term with (21) and we have

$$R_T^{delay} \geq \sum_{k > M} \frac{(1 - o(1))\log(T)}{2\theta\Delta_k} + \left( \mathbb{E}[d] - \sigma_d\sqrt{\frac{\theta}{1-\theta}} \right) \frac{M}{2K} \sum_{k > M} \Delta_k - \frac{2}{\theta}.$$

