# OpenReview forum: "Multi-player Multi-armed Bandits with Delayed Feedback"
_ICLR.cc/2025/Conference — Submitted to ICLR 2025_

### Official Review · Reviewer_j5FD · 2024-10-18

**Soundness:** 2
**Presentation:** 3
**Contribution:** 2
**Rating:** 5
**Confidence:** 4

**Summary:**

This paper studies multi-player multi-armed bandit (MMAB) problem with delayed feedback, motivated by the application of cognitive radio networks. Unlike previous MMAB problems that assume instantaneous feedback from arms, this work tackles the challenge posed by delayed feedback. To overcome this challenge, this work proposes a decentralized delayed successive elimination (DDSE) algorithm, which operates in three stages: exploration, communication, and exploitation. The proposed DDSE algorithm enables players to adapt to delayed feedback and avoid collision. This work theoretically analyzes the upper bound of regret for the DDSE algorithm and further compares the regret with two benchmark cases: DDSE without delay estimation and centralized lower bound. By comparison, it shows that the DDSE achieves a near-optimal regret bound.

**Strengths:**

1.	The problem of MMAB with delayed feedback is well-motivated and highly relevant to real-world applications.
2.	Introducing delayed feedback significantly increases the complexity of the already challenging MAB problem. The authors effectively decompose the regret to handle this complexity and present solid theoretical results.
3.	The paper is well-written and easy to follow.

**Weaknesses:**

1.	My main concern lies on the ID allocation for the leader-follower structure in the DDSE algorithm. If the central planner can assign an ID to each player, this DDSE algorithm is no longer fully decentralized. In many cognitive radio networks, sensing nodes are dynamic, and some nodes are even hidden or unknown to the network operator.
2.	The communication assumption weakens the solution in this work.
3.	I suggest the authors to move Subsection 5.3 ahead of Subsection 5.1 for better logic, as the centralized lower bound serve as the benchmark.
4.	In the experiments, the number of players $M$ is a relatively small compared to typical application of cognitive radio networks.
5.	In the experiments, the authors simply compare DDSE with two methods that do not account for delay. This comparison may be somewhat unfair. If there is no other available algorithm, it would be better to compare DDSE with the benchmark centralized algorithm.

**Questions:**

1.	In cognitive radio networks, players are usually dynamic. Additionally, there are some hidden nodes (players) that are unknown to each other. In this case, will the DDSE algorithm still work?
2.	If a player $j$ is waiting for the feedback from arm $k$ (i.e., $t<s+d_s^j$) and another player $l$ pulls this arm $k$, will there be a collision? If a collision occurs, will player $j$ fail to obtain a reward from arm $k$ after waiting $t-s$ time slots?

---

> ### Author Response · Authors · 2024-11-22
> **Response to Reviewer j5FD (Part 1)**
>
> Thank you for your comments about our manuscript. We have studied the comments carefully, and find them valuable for improving our paper. The responses to your comments are as follows:
> # Response to Weakness 1:
>  *Q: This DDSE algorithm is not fully decentralized. In many cognitive radio networks, sensing nodes are dynamic, and some nodes are even hidden or unknown to the network operator.*
>
> We have considered a fully decentralized setting at the beginning of our work. In such fully decentralized multi-player bandits, players do not know the existence of others so there is no guarantee to avoid collisions. Thus, decentralized MMAB algorithms need an initialization phase where players utilize collisions to get some information before exploration, so that they can select arms in a round-robin way. There are two kinds of methods for initialization.
>
> 1. [1, 2, 3] adopt a musical chair method, where each player preempts an arm until no collision happens. After preempting an arm, players then intentionally select some arms. By counting the times of accumulated collision in this period, each player can know $M$ and her ID among these players.
>
> 2. In [4], players also try to preempt arms once. If they fail to preempt it, i.e., receive a collision, they will select specific arms later to pass the information that she has not found her arm to other players. When no collision occurs, it means that everyone finds the proper arm. Then they perform a procedure similar to [1, 2, 3] and get the information on $M$ and their ID.
>
> However, when delay is introduced to the environment, players can not receive the collision immediately. If the delay is too long, players must wait in the initialization phase to receive the feedback of collision, because they need to get $M$ and their ID to start an exploration where they can select arms in a round-robin way. Actually, players always hope that they should receive some feedback in a certain known period, which conflicts with the scenario where feedback is delayed for an unknown period.
>
> Of course, if we assume that delay is bounded by a known value $d _ {\max}$, the problem of initialization will be solved easily by using the classical technique and waiting for extra $d _ {\max}$ rounds in the beginning. In cognitive radio networks, transmission delay is sometimes bounded by protocol limits, so it is also natural to consider a bounded delay where we can sue $d _ {\max}$ as the input of algorithms to perform an initialization.
>
> Our assumption on delay is more mild than this bounded delay. We allow some exceedingly large and unknown delay in the environment. If the assumption on the initialization does not exist, i.e., players are fully decentralized, we can also implement that pre-mentioned technique in our algorithms. For example, players start with an initialization phase. Although their feedback might not be completed, they still begin to explore arms. If a player in exploration has a collision that should not have happened, she notifies others also by sending collisions at a specific time. After a period of delay, players receive the signal and start an initialization again. Then comes the next exploration. By doing so, it is possible to give a probability guarantee using our sub-Gaussian delay assumption. However, it will make the notation more complicated. The existing problem is enough challenging and our work is the first paper handling delay in multi-player bandits where a collision occurs when selecting the same arm. Our goal is to pave the way for further study in this area. A fully decentralized MMAB can be discussed in future work.
>
> # Response to Weakness 2:
> *Q: The communication assumption weakens the solution in this work.*
>
> We do not assume that players can freely communicate. They can only know $M$ and their own ID. The communication phase is designed to solve the difficulty that player can not directly communicate. We review that a player $j$ receives feedback on the reward of the arm that she selected several rounds ago and whether she has a collision on the arm. The collision indicates that there exists at least another player who also selected the same arm. In our algorithm design, collisions that happen on specific periods represent some implicit information. Thus, by intentionally sending collisions, players can pass some information to others.
>
> Next, we explain our communication phase here. The primary goal of communication phase is to pass the update of $\mathcal{M} _ p^j$.  Each communication phase is composed of three blocks. We consider $M=4$ and $K=6$. The example is:
>
> \begin{aligned}
> \mathrm{Leader:} \ &\mathcal{M}^4_1 = \\{4, 2, 6, 3\\}, \mathcal{M}^4_2 = \\{4, 2, 6, 1\\}, \mathcal{M}^4_3 = \\{4, 2, 6, 5\\} \\\\
> \mathrm{Follower\ 1:} \ &\mathcal{M}^1_1 = \\{4, 2, 6, 3\\} \\\\
> \mathrm{Follower\ 2:} \ &\mathcal{M}^2_1 = \\{4, 2, 6, 3\\} \\\\
> \mathrm{Follower\ 3:} \ &\mathcal{M}^3_1 = \\{4, 2, 6, 3\\}
> \end{aligned}

---

> > ### Comment · Reviewer_j5FD · 2024-11-26
> >
> > I thank the authors' response. However, my primary concern remains unresolved. Specifically, in practice, ID allocation by the leader is unrealistic since not all nodes are known in advance. Furthermore, this ID allocation significantly undermines the contribution of the distributed algorithm design.

---

> > > ### Author Response · Authors · 2024-12-03
> > >
> > > Thanks for your feedback. We will improve it in next version.

---

> ### Author Response · Authors · 2024-11-22
> **Response to Reviewer j5FD (Part 2)**
>
> ## The first block is used for removing an arm from $\mathcal{M}^j _ p$ .
> The leader first finds the bad arm $a^- _ p$ in $\mathcal{M}^j _ p$ and identifies the position of $a^- _ p$ . Then the leader selects the $i _ {a^- _ p}$ -th arm in $\mathcal{M}^j _ {p-q _ j}$ for $M$ times. Followers still select arms in $\mathcal{M}^j _ {p-q _ j}$ in a round-robin way. After a collision occurs, followers save the position that a collision happens. Then when she gets both the position of the arm to be removed and the arm to be added, she will update the best arm set.
> In the example, players in phase $2$ (i.e., $p=2$) but they still use $\mathcal{M} _ 1^j$. $t _ 1$ to $t _ 4$ denotes the rounds in the first block. We suppose that the leader wants to remove arm $1$ from $\mathcal{M} _ 2^4$. The arm selection is in the table.
>
> |            | $t _ 1$ | $t _ 2$ | $t _ 3$ | $t _ 4$ |
> | ---------- | ----- | ----- | ----- | ----- |
> | Leader     | 3     | 3     | 3     | 3     |
> | Follower 1 | 4     | 2     | 6     | 3     |
> | Follower 2 | 2     | 6     | 3     | 4     |
> | Follower 3 | 6     | 3     | 4     | 2     |
>
> Each follower collides on arm $3$ once. They remember the position of collision in $p=2$ is $4$.
>
> ## The second block is used for adding an arm from $\mathcal{M}^j _ p$.
> The leader finds a better arm that has higher empirical rewards but not in $\mathcal{M}^j _ p$. Then she wants to pass the new arm to followers. Because the new arm $a^+ _ p$ might not be in $\mathcal{M}^j _ p$ or $\mathcal{M}^j _ {p-q _ j}$, we can not let followers receive the collision information via the best arm set. Thus, we utilize the whole arm set. The leader continuously selects $a^+ _ p$ for $K$ times. Followers select all arms in a round-robin way. This block continues $K$ times because the length of the original arm is $K$ and we hope to use the original whole arm set to pass the information.
>
> In the example, $t _ 5$ to $t _ 6$ denotes the rounds in the second block. We suppose that the leader wants to add arm $1$.
>
> |            | $t _ 5$ | $t _ 6$ | $t _ 7$ | $t _ 8$ | $t _ 9$ | $t _ {10}$ |
> | ---------- | ----- | ----- | ----- | ----- | ----- | -------- |
> | Leader     | 5     | 5     | 5     | 5     | 5     | 5        |
> | Follower 1 | 1     | 2     | 3     | 4     | 5     | 6        |
> | Follower 2 | 2     | 3     | 4     | 5     | 6     | 1        |
> | Follower 3 | 3     | 4     | 5     | 6     | 1     | 2        |
>
> Each follower collides on arm $5$ once. They remember that the arm where the collision happens is $5$.
> ## The third block is used for passing the ending of exploration.
> Leader explores and gradually eliminates all sub-optimal arms from $[K]$. However, as for followers, they do not know when the exploration of the leader ends. If the leader's exploration has ended and passes all optimal arms to followers, continuing to enter the communication phase is a waste of time for followers. Therefore, a block used for passing the ending of exploration is necessary. Players' actions are similar to the first block.
> ## After a period of delay:
> Note that the collisions in the first and second blocks can not be received immediately. After a period of delay, follower $j$ receives the position $4$ and the arm to be added is $5$. She also knows the feedback is from phase $2$. Therefore, she updates:
> $$
> \mathcal{M} _ 1^j = \\{4, 2, 6, 3\\} \rightarrow \mathrm{replace\ the\ arm\ in\ position\ 4\ to\ arm\ 5} \rightarrow\mathcal{M}^j _ 2 = \\{4, 2, 6, 5\\}
> $$
> No matter how late the feedback is received by the follower, she can update the correct $\mathcal{M} _ p^j$.
>
> # Response to Weakness 3:
> *Q: I suggest the authors to move Subsection 5.3 ahead of Subsection 5.1 for better logic, as the centralized lower bound serve as the benchmark.*
>
> We sincerely thank the reviewer for the valuable suggestion regarding the organization of the theoretical analysis section. In response, we have moved the lower bound to the beginning of this section, as it serves as a natural benchmark for the subsequent analysis.
> # Response to Weakness 4:
>
> *Q: In the experiments, the number of players $M$ is a relatively small compared to typical application of cognitive radio networks.*
>
> We truly appreciate that the reviewer points out the concern regarding the relatively small number of players. To address this, we have conducted additional experiments with larger parameters, specifically for $M=30$ and $M=40$. The results, which are now included in Appendix B, demonstrate that our algorithms continue to outperform others as the number of players increases.

---

> ### Author Response · Authors · 2024-11-22
> **Response to Reviewer j5FD (Part 3)**
>
> # Response to Weakness 5:
> *Q: In the experiments, the authors simply compare DDSE with two methods that do not account for delay. This comparison may be somewhat unfair. If there is no other available algorithm, it would be better to compare DDSE with the benchmark centralized algorithm.*
>
> As Reviewer N47W suggests, we have compared our algorithms with [5, 6, 7]. All the experimental results show that our algorithms perform best. We also compare DDSE in both decentralized and centralized settings. Experimental results in Figure 10 show that the performance of DDSE closely matches that in the centralized setting.
>
> # Response to Question 1:
> *Q: In cognitive radio networks, players are usually dynamic. Additionally, there are some hidden nodes (players) that are unknown to each other. In this case, will the DDSE algorithm still work?*
>
> ## Dynamic Players:
> If players are dynamic, the setting is called asynchronous multi-player bandits. In this setting, players can enter the game and leave at any time.  The latest study on asynchronous MMAB is [8]. However, [8] relies on several assumptions:
>
> 1. There exists a centralized environment, where players can freely communicate with others.
>
> 2. Although players enter the game at different times, they leave at the same time $T$.
>
> 3. At each time, the probability that every player enters the game is the same and known.
>
> The area of centralized asynchronous MMAB without assumptions is still blank, let alone decentralized asynchronous MMAB. In decentralized bandits, delay has already significantly complicated the problem because players rely on the feedback of collisions to get some information from other players. Incomplete feedback causes inconsistency between players, leading to staggered exploration, frequent collisions, or premature exploitation.
>
> ## Hidden Players:
> As stated in "Response to Weakness 1", if players have no prior knowledge about others, an initialization phase can be introduced. Players start with an initialization phase where, despite incomplete feedback, they begin to explore arms. If a collision occurs during exploration that should not have happened, the affected player notifies others by sending a collision signal at a specific time. After a period of delay, players receive the signal and re-enter the initialization phase, followed by the next round of exploration. By adopting this approach, it is possible to provide probabilistic guarantees under our sub-Gaussian delay assumption. However, this would complicate the notation significantly. The problem addressed in this work is already sufficiently challenging, as our study is the first to tackle delays in multi-player bandits where collisions occur when players select the same arm. Our goal is to pave the way for further study in this area. A fully decentralized MMAB can be discussed in future work.
>
> # Response to Question 2:
> *Q: If a player $j$ is waiting for the feedback from arm $k$ (i.e., $t<s+d _ s^j$) and another player $l$ pulls this arm $k$, will there be a collision? If a collision occurs, will player $j$ fail to obtain a reward from arm $k$ after waiting $t−s$ time slots?*
>
> In our paper, players selecting the same arm at the same time is called "collision". A player $j$ selects arm $k$ at time $s$. Another player $\ell$  also selects arm $k$ at time $s$.  Then we call that the two players collide with each other, but they do not receive the feedback immediately. The delay $d _ s^j$ and $d _ s^{\ell}$ do not have to be the same. Player $j$ and player $\ell$ will receive their feedback that they have collided at time $s$ in different time slots. When the feedback from time $s$ has not been received, player $j$ can wait for the feedback from arm $k$ (i.e., $t<s+d _ s^j$) while selecting other arms. "Waiting" does not mean that the player is idle. If another player $\ell$ selects arm $k$ also at time $s$, then both player $j$ and $\ell$ will receive collisions. If the player $\ell$ selects arm $k$ at $t^{\prime}$ (We define $s \leq t^{\prime} \leq s+d _ s^j$. ), player $j$ will not receive a collision at time $s+d _ s^j$, because at time $s$, only $j$ selects the arm $k$.

---

> ### Author Response · Authors · 2024-11-22
> **Response to Reviewer j5FD (Part 4)**
>
> # Reference
> [1] Boursier, Etienne, and Vianney Perchet. "SIC-MMAB: Synchronisation involves communication in multiplayer multi-armed bandits." _Advances in Neural Information Processing Systems_ 32 (2019).
>
> [2] Shi, Chengshuai, et al. "Decentralized multi-player multi-armed bandits with no collision information." _International Conference on Artificial Intelligence and Statistics_. PMLR, 2020.
>
> [3] Huang, Wei, Richard Combes, and Cindy Trinh. "Towards optimal algorithms for multi-player bandits without collision sensing information." _Conference on Learning Theory_. PMLR, 2022.
>
> [4] Wang, Po-An, et al. "Optimal algorithms for multiplayer multi-armed bandits." _International Conference on Artificial Intelligence and Statistics_. PMLR, 2020.
>
> [5] Besson, Lilian, and Emilie Kaufmann. "Multi-player bandits revisited." _Algorithmic Learning Theory_. PMLR, 2018.
>
> [6] Bistritz, Ilai, and Amir Leshem. "Distributed multi-player bandits-a game of thrones approach." _Advances in Neural Information Processing Systems_ 31 (2018).
>
> [7] Tibrewal, Harshvardhan, et al. "Distributed learning and optimal assignment in multiplayer heterogeneous networks." _IEEE INFOCOM 2019-IEEE Conference on Computer Communications_. IEEE, 2019.
>
> [8] Richard, Hugo, Etienne Boursier, and Vianney Perchet. "Constant or Logarithmic Regret in Asynchronous Multiplayer Bandits with Limited Communication." _International Conference on Artificial Intelligence and Statistics_. PMLR, 2024.

---

### Official Review · Reviewer_ix9S · 2024-11-01

**Soundness:** 3
**Presentation:** 2
**Contribution:** 3
**Rating:** 8
**Confidence:** 3

**Summary:**

This paper considers a multi-arm multi-player bandit setup with delayed reward. The paper proposes novel algorithms to counter the delay in receiving the reward. The paper bounds the regret in the decentralized setting.

**Strengths:**

1. Bounding the regret in the multi-armed multi-agent bandit setup is challenging. The paper additionally consider the delay, hence, the contribution seems to be significant.

2. The paper achieves the regret bound.

3. Empirical results show the efficacy of the proposed approach.

Post-rebuttal Edit: Addressing the delay parameter in the multi-armed collision model is important. Thus, I am happy to accept this paper.

**Weaknesses:**

1. The paper considers cognitive radio setup. However, cognitive radio is hardly used in practice, it is only of academic interest. Can the paper provide any other relevant examples?

2. The paper is very hard to read, hence, the contributions are obscure.

**Questions:**

1. Can the authors highlight the main technical challenges? Delay in the multi-armed setting is considered, while the reviewer agrees that the collision model does complicate things, in the technical level, how the analysis will be different is not clear.

---

> ### Author Response · Authors · 2024-11-22
> **Response to Reviewer ix9S (Part 1)**
>
> Thank you for your comments about our manuscript. We have studied the comments carefully, and find them valuable for improving our paper. The responses to your comments are as follows:
> # Response to Weakness 1:
> *Q: Can the paper provide any other relevant examples except cognitive radio?*
>
> ## Autonomous Vehicles in Traffic Management:
> When multiple autonomous vehicles (players) choose the same lane or intersection (arm) at the same time, traffic congestion or collisions can occur. By applying our algorithm, route optimization and intersection management can be significantly improved, leading to fewer collisions and smoother traffic flow.
> ## Resource Scheduling in Cloud Computing:
> In cloud computing, users (players) often compete for access to the same virtual machines or servers (arms), which can create resource bottlenecks and degrade performance. Our algorithm dynamically allocates tasks to available resources, effectively reducing conflicts and improving system efficiency.
> # Response to Weakness 2:
> *Q: The paper is very hard to read, hence, the contributions are obscure.*
>
> Thank you sincerely for the thoughtful comments that clarity could be improved. We have revised our manuscript by refining notations, enhancing the descriptions of the algorithms, and improving the presentation of the proofs. Below, we summarize our main contributions for better clarity:
> 1. We propose a novel framework for multi-player multi-armed bandits (MMAB) with delayed feedback. To the best of our knowledge, we are the first to address delay in MMAB settings where selecting the same arm results in collisions.
>
> 2. To tackle this challenge, we introduce the DDSE (Decentralized Delayed Successive Elimination) algorithm. In DDSE, players coordinate to utilize the same best empirical arm set, determined based on their delay estimations, before each exploration phase. This ensures that collisions are effectively avoided.
>
> 3. We derive a regret bound of DDSE, and study the regret bound of DDSE in centralized setting. We also derive a centralized lower bound in MMAB with delayed feedback. Compared with the lower bound, the regret of DDSE is near-optimal.
>
> 4. Finally, we validate the efficacy of DDSE through numerical experiments conducted on both synthetic and real-world datasets.
>
> We describe our main algorithm and its proof here.
> ## Algorithm: DDSE
> Our algorithm is composed of exploration and communication. Because the environment is decentralized, we design this communication phase where players can pass implicit information using intentional collisions to simulate communication. The player with ID $M$ becomes the leader and others are followers.
>
> **Exploration**
>
> In exploration, the leader explores all active arms while followers only select arms from a specific best arm set. The set of followers is updated when they receive information from the leader in communication phase. Due to the decentralized environment, each player may have a different best arm set. Thus, we define $\mathcal{M} _ p^j$ as the best arm set of player $j$ at phase $p$. Note that the leader sends the information to followers only after she knows the results, but the information that followers receive is delayed for an unknown time. Thus, after every communication phase ends, the updated $\mathcal{M} _ p^M$ is different with $\mathcal{M} _ p^j, j<M$. Different best arm sets lead to: (1) collision in the exploration phase, (2) passing wrong information in the communication phase. To avoid this situation, we introduce our method of delay estimation. The intuition is, that although the current best arm set is different, players can select arms in the previous best arm set which has been received completely. Lemma 1 and Lemma 2 guarantee that players select the same best arm set so that no collision occurs in exploration phase.
>
> **Communication**
>
> The goal of communication phase is to pass the update of $\mathcal{M} _ p^j$.  Each communication phase is composed of three blocks. We consider an example with $M=4$ and $K=6$ to better understand the algorithm. We suppose that players are in phase $2$. They use the previous arm set $\mathcal{M} _ 1^j$. The leader wants to update to $\mathcal{M} _ 3^4$.
>
> \begin{aligned}
> \mathrm{Leader:} \ &\mathcal{M}^4_1 = \\{4, 2, 6, 3\\}, \mathcal{M}^4_2 = \\{4, 2, 6, 1\\}, \mathcal{M}^4_3 = \\{4, 2, 6, 5\\} \\\\
> \mathrm{Follower\ 1:} \ &\mathcal{M}^1_1 = \\{4, 2, 6, 3\\} \\\\
> \mathrm{Follower\ 2:} \ &\mathcal{M}^2_1 = \\{4, 2, 6, 3\\} \\\\
> \mathrm{Follower\ 3:} \ &\mathcal{M}^3_1 = \\{4, 2, 6, 3\\}
> \end{aligned}

---

> ### Author Response · Authors · 2024-11-22
> **Response to Reviewer ix9S (Part 2)**
>
> **1. The first block is used for removing an arm from $\mathcal{M}^j _ p$ .**
>
> The leader first finds the bad arm $a^- _ p$ in $\mathcal{M}^j _ p$ and identifies the position of $a^- _ p$ . Then the leader selects the $i _ {a^- _ p}$ -th arm in $\mathcal{M}^j _ {p-q _ j}$ for $M$ times. Followers still select arms in $\mathcal{M}^j _ {p-q _ j}$ in a round-robin way. After a collision occurs, followers save the position that a collision happens. Then when she gets both the position of the arm to be removed and the arm to be added, she will update the best arm set.
> In the example, players in phase $2$ (i.e., $p=2$) but they still use $\mathcal{M} _ 1^j$. $t _ 1$ to $t _ 4$ denotes the rounds in the first block. We suppose that the leader wants to remove arm $1$ from $\mathcal{M} _ 2^4$. The arm selection is in the table.
>
> |            | $t _ 1$ | $t _ 2$ | $t _ 3$ | $t _ 4$ |
> | ---------- | ----- | ----- | ----- | ----- |
> | Leader     | 3     | 3     | 3     | 3     |
> | Follower 1 | 4     | 2     | 6     | 3     |
> | Follower 2 | 2     | 6     | 3     | 4     |
> | Follower 3 | 6     | 3     | 4     | 2     |
>
> Each follower collides on arm $3$ once. They remember the position of collision in $p=2$ is $4$.
>
> **2. The second block is used for adding an arm from $\mathcal{M}^j _ p$.**
>
> The leader finds a better arm that has higher empirical rewards but not in $\mathcal{M}^j _ p$. Then she wants to pass the new arm to followers. Because the new arm $a^+ _ p$ might not be in $\mathcal{M}^j _ p$ or $\mathcal{M}^j _ {p-q _ j}$, we can not let followers receive the collision information via the best arm set. Thus, we utilize the whole arm set. The leader continuously selects $a^+ _ p$ for $K$ times. Followers select all arms in a round-robin way. This block continues $K$ times because the length of the original arm is $K$ and we hope to use the original whole arm set to pass the information.
> In the example, $t _ 5$ to $t _ 6$ denotes the rounds in the second block. We suppose that the leader wants to add arm $1$.
>
> |            | $t _ 5$ | $t _ 6$ | $t _ 7$ | $t _ 8$ | $t _ 9$ | $t _ {10}$ |
> | ---------- | ----- | ----- | ----- | ----- | ----- | -------- |
> | Leader     | 5     | 5     | 5     | 5     | 5     | 5        |
> | Follower 1 | 1     | 2     | 3     | 4     | 5     | 6        |
> | Follower 2 | 2     | 3     | 4     | 5     | 6     | 1        |
> | Follower 3 | 3     | 4     | 5     | 6     | 1     | 2        |
>
> Each follower collides on arm $5$ once. They remember that the collision happens on arm $5$.
>
> **3. The third block is used for passing the ending of exploration.**
>
> Leader explores and gradually eliminates all sub-optimal arms from $[K]$. However, as for followers, they do not know when the exploration of the leader ends. If the leader's exploration has ended and passes all optimal arms to followers, continuing to enter the communication phase is a waste of time for followers. Therefore, a block used for passing the ending of exploration is necessary. Players' actions are similar to the first block.
>
> **4. After a period of delay:**
>
> Note that the collisions in the first and second blocks can not be received immediately. After a period of delay, follower $j$ receives the position $4$ and the arm to be added is $5$. She also knows the feedback is from phase $2$. Therefore, she updates:
> $$
> \mathcal{M}_1^j = \\{4, 2, 6, 3\\} \rightarrow \mathrm{replace\ the\ arm\ in\ position\ 4\ to\ arm\ 5} \rightarrow\mathcal{M}^j_2 = \\{4, 2, 6, 5\\}
> $$
> Therefore, although players use the previous arm set $\mathcal{M} _ 1^j$, followers can still receive the correct update information of $\mathcal{M} _ 2^j$ even though they do not know $\mathcal{M} _ 2^j$.

---

> ### Author Response · Authors · 2024-11-22
> **Response to Reviewer ix9S (Part 3)**
>
> ## Regret Analysis of DDSE
> **Exploration Phase**
>
> Lemma 5 ensures that the delayed feedback from the communication phase of all followers is bounded. Then Lemma 6 establishes the accuracy of the estimates for $\mathbb{E}[d]$ and $\sigma _ d^2$. As a result, player $j$ can correctly determine $q _ j$ and align with the same best empirical arm set, thereby preventing collisions caused by inconsistencies between the leader and the followers. Thus, the regret in exploration phase is generated from (1) selecting sub-optimal arms, (2) players not receiving any feedback initially, and (3) the leader not entering the exploitation phase immediately after identifying all sub-optimal arms. During the period after the leader identifies all sub-optimal arms but before entering the exploitation phase, the leader still needs to maintain consistency with followers by selecting arms in $\mathcal{M} _ {p-q _ M}$. We separately bound these terms in Appendix D.
>
> **Communication Phase**
>
> We have already known that the length of each communication phase is $K+2M$. Note that players enter a communication phase every $KM\log(T)$ rounds. The next step is to bound the times that the leader needs to receive feedback and eliminate all sub-optimal arms. Thus, $T _ {expl}/KM\log(T)$ indicates the number of times players enter a communication phase, which we then multiply by the phase length $K+2M$. Note that there are $M$ players in communication phase, so we finally take a union bound of $M$.
>
> Combining the results in exploration and communication phase, we derive the regret bound of DDSE. By comparing with the lower bound in Theorem 1, DDSE is near-optimal.
>
> # Response to Question 1:
> *Q: Can the authors highlight the main technical challenges?*
>
> We appreciate the reviewer's insightful question. The main technical challenge lies in accurately estimating the delay within the decentralized multi-armed bandit setting, particularly under the collision model. While delay analysis has been explored previously, our approach introduces a novel consideration of sub-Gaussian delays, which adds complexity to bounding delays during the communication phase. Specifically, we address the challenge by proving the correctness of our estimations for both  $\mathbb{E}[d]$ and $\sigma^2 _ d$  (Lemma 2) and analyzing potential errors in delay estimation. Leveraging the fact that the square of a sub-Gaussian variable is sub-Exponential, we rigorously bound the probabilities of deviations $|\hat{\mu} _ {d _ t^j}-\mathbb{E}[d]| \leq \epsilon _ {\mu}$ and $|\hat{\sigma} _ {d _ t^j}^2-\sigma _ d^2| \leq \epsilon _ {\sigma}$. In Lemma 9, applying the inverse Jensen inequality twice allows us to derive $\mathbb{E}[s _ p-s _ {p-1}]$, so that we can utilize $\mathbb{E}[s _ p-s _ {p-1}]=KM\log(T)$. Combining this with the term $T$ on the outside, we finally get the bound of $R_{\mathcal{F}}$.

---

> > ### Comment · Reviewer_ix9S · 2024-11-25
> > **Thank you**
> >
> > I thank the authors for answering my questions. I have some follow-up questions--
> >
> > 1. How do the agents select the leader? Is there any communication signal sent to agree on the leader?
> >
> > 2. In Algorithm 1, line 3, it says `explore', how does the leader explore here? Is it like uniformly picking any of the arms?
> >
> > 3. Since it is a collision model and the rewards may be different for different players, how do they select the optimal set of arms? I mean consider the following scenario where players 1 and 2 have the best reward for arm 1, but poor rewards for arm 2, then what should be the solution to it? Now if the number of players and arms increases, it seems that the original problem becomes a combinatorial problem in choosing the optimal arms. That begs the question about the practicality of this setup apart from the academic interest. In wireless communication, we do time-division multiplexing to avoid collision (same for the traffic intersection problem that you describe).
> >
> > 4.  I am a little bit confused about $\Delta_k$, here, should not the optimal arm and thus the optimality gap depends on the individual player? Is there an inherent assumption that the rewards are the same for the players?

---

> > > ### Author Response · Authors · 2024-11-25
> > > **Response to Follow-up Questions**
> > >
> > > Thank you very much for your thoughtful and detailed feedback. We sincerely appreciate the time and effort you have taken to review our work and provide valuable questions.
> > > # Response to Question 1
> > > We assume that players are initialized with their own ID. The player with ID $M$ becomes the leader and others are followers.
> > > # Response to Question 2
> > > Yes, the leader uniformly picks arms from the active arm set $\mathcal{K}$. Specifically, she first pulls arms in the set of best empirical arms with followers. Then she selects other arms in $\mathcal{K}$ in a round-robin way while skipping arms in the best arm set. We also use an example to explain the process. Let $K=8$ and $M=4$:
> > >
> > > \begin{aligned}
> > > \mathrm{Leader:} \ &\mathcal{M}^4 _ 1 = \\{4, 2, 6, 3\\} \quad \mathrm{to} \quad \mathcal{M}^4_2 = \\{4, 2, 6, 8\\} \\\\
> > > \mathrm{Follower\ 1:} \ &\mathcal{M}^1 _ 1 = \\{4, 2, 6, 3\\} \\\\
> > > \mathrm{Follower\ 2:} \ &\mathcal{M}^2 _ 1 = \\{4, 2, 6, 3\\} \\\\
> > > \mathrm{Follower\ 3:} \ &\mathcal{M}^3 _ 1 = \\{4, 2, 6, 3\\}
> > > \end{aligned}
> > >
> > > In this example, we suppose that players are focusing on $\mathcal{M} _ 1^j$. They select arms as follows:
> > >
> > > | Player     | $t_1$ | $t_2$ | $t_3$ | $t_4$ | $t_5$ | $t_6$ | $t_7$ | $t_8$ |
> > > | ---------- | ----- | ----- | ----- | ----- | ----- | ----- | ----- | ----- |
> > > | Leader     | 3     | 4     | 2     | 6     | 1     | 5     | 7     | 8     |
> > > | Follower 1 | 4     | 2     | 6     | 3     | 4     | 2     | 6     | 3     |
> > > | Follower 2 | 2     | 6     | 3     | 4     | 2     | 6     | 3     | 4     |
> > > | Follower 3 | 6     | 3     | 4     | 2     | 6     | 3     | 4     | 2     |
> > >
> > > The leader first selects arms in $\mathcal{M} _ 1^4$ with followers from $t _ 1$ to $t _ 4$. After pulling all arms in $\mathcal{M} _ 1^4$, she begins to select other arms in $\mathcal{K}\backslash \mathcal{M} _ 1^4$. The arm selection in one exploration continues for $KM\log(T)$ times. Note that the leader gradually eliminates sub-optimal arms, so $\mathcal{K}$ is shrinking.
> > >
> > > # Response to Question 3
> > > We consider the scenario of $K \geq M$ which ensures that every player can find at least one arm without collision. Since there are $M$ players, we have $M$ optimal arms and $K-M$ sub-optimal arms. Our goal is to minimize regret which is defined as:
> > >
> > > $$
> > > R _ T:= T\sum _ {j\in[M]}\mu _ {(j)} - \mathbb{E}\left[ \sum _ {t=1}^{T} \sum _ {j\in[M]} r^j(t) \right],
> > > $$
> > >
> > > where $\mu _ {(j)}$ is $j$-th order statistics of $\mu$, i.e., $\mu _ {(1)} \geq \mu _ {(2)} \geq ... \geq \mu _ {(K)}$. In other words, $R _ T$ is the accumulated regret of all players. The reward expectation $\mu _ {(k)}$ of arm $k$ is the same for different players. The optimal solution is that players select the first $M$ optimal arms in a staggered manner.  Within these first $M$ optimal arms, it does not matter which arm each player chooses, as long as they do not collide.
> > >
> > > In the scenario, if we only have two arms, players $1$ and $2$ select these two arms separately because they do not have other choices. When $K>2$, if arm $2$ is optimal (its reward expectation is lower than arm $1$ but higher than other arms), players $1$ and $2$ select arm $1$ and $2$, or arm $2$ and $1$. If arm $2$ is not an optimal arm, players should find other optimal arms.
> > >
> > > # Response to Question 4
> > > As said in response to Q3, the reward expectation $\mu _ {(k)}$ of arm $k$ is the same for different players, so $\Delta _ k$ is dependent on arm $k$.
> > >
> > > Thank you again for your insightful questions. Please feel free to reach out with any further queries or suggestions.

---

> > > > ### Comment · Reviewer_ix9S · 2024-11-25
> > > > **Thanks again**
> > > >
> > > > Thank you for answering my follow-up questions. It indeed clarified my doubt on how to find the optimal set of arms in case the rewards are different for different players. However, here the setting is that the rewards are the same across the players.
> > > >
> > > > Two more questions-- Regret is generally associated with the high-probability bound, however, I did not see those effects in Theorems 1 and 2, and we do not see this effect. Is it an average regret?
> > > >
> > > > Do the authors have a lower bound result on the delay-distribution parameters to show the tightness of the result?

---

> > > > > ### Author Response · Authors · 2024-11-26
> > > > >
> > > > > We truly appreciate your thoughtful questions. It has allowed us to clarify and elaborate on key aspects of our results.
> > > > > # Response to Question 1
> > > > > Yes, it is expected regret. The regret is defined as:
> > > > >
> > > > > $$
> > > > > R _ T:= T\sum _ {j\in[M]}\mu _ {(j)} - \mathbb{E}\left[ \sum _ {t=1}^{T} \sum _ {j\in[M]} r^j(t) \right],
> > > > > $$
> > > > >
> > > > > where the expectation is taken over the randomness of the rewards.
> > > > >
> > > > > # Response to Question 2
> > > > > The centralized lower bound is provided in Theorem 1. We recall that the lower bound is:
> > > > >
> > > > > $$
> > > > > R _ T \geq \underbrace{\sum _ {k>M}\frac{(1-o(1))\log(T)}{2\theta\Delta _ k}} _ {\mathrm{term\ I}} + \underbrace{\left(\mathbb{E}[d] - \sigma _ d\sqrt{\frac{\theta}{1-\theta}}\right) \frac{M}{2K}\sum _ {k>M}\Delta _ k  - \frac{2}{\theta}} _ {\mathrm{term\ II}} , \tag*{(1)}
> > > > > $$
> > > > >
> > > > > where $\mathbb{E}[d]$ is the expectation of a delay distribution and $\sigma _ d^2$ is the sub-Gaussian parameter. Define $d(\theta):=\min\{ \gamma\in\mathbb{N}|P(d \leq \gamma) \geq \theta \}$ as the quantile function of the delay distribution, so $\theta$ in the lower bound is a quantile.
> > > > >
> > > > > The regret bound of DDSE is in Theorem 2. We also rewrite here:
> > > > > $$
> > > > > \begin{aligned}
> > > > > 	R _ {T} \leq &\sum _ {k>M}\frac{323\log(T)}{\theta \Delta _ k} +  \left(9 +\frac{2M\sum _ {k>M}\Delta _ k}{K-M}\right)\mathbb{E}[d] + \sigma _ d \left(3\sqrt{6} + 6\sqrt{2\log(\frac{1}{1-\theta})}\right) \\\\
> > > > > 	&+ \frac{\sigma _ d M}{K-M}\sum _ {k>M}\Delta _ k\sqrt{\log\left((M-1)(K+2M)\right)}  + C_1, \\\\
> > > > > 	\leq & \underbrace{\sum _ {k>M}\frac{323\log(T)}{\theta \Delta_k}} _ {\mathrm{term\ A}} + \underbrace{\left( 2\mathbb{E}[d]+\sigma _ d\sqrt{3\log(K)} \right)\frac{M}{K-M}\sum _ {k>M}\Delta _ k} _ {\mathrm{term\ B}} + \underbrace{\left( 9\mathbb{E}[d]+6\sqrt{2\log(\frac{1}{1-\theta})} \right)} _ {\mathrm{term\ C}} \\\\
> > > > > 	& + \underbrace{3\sqrt{6}\sigma_d} _ {\mathrm{term\ D}} + \underbrace{C _ 1} _ {\mathrm{term\ E}}  \quad \quad \quad \quad \quad \quad \quad \quad \quad \quad \quad \quad \quad \quad \quad \quad \quad \quad \quad \quad \quad \quad \quad \quad \quad \quad \quad \quad (2)
> > > > > \end{aligned}
> > > > > $$
> > > > > where $C _ 1=  \sum _ {k>M}\frac{195}{\theta\Delta _ k^2} + \frac{4Me^{-\delta^2/2}}{\delta^2}$.
> > > > >
> > > > > Only the first terms in (1) and (2) are related to $T$. Term A is aligned with term I up to constant factors. Term E arises due to the decentralized environment and is not related to delay. Regarding delay, a comparison of term II with terms B, C, and D reveals that the difference on $K$ and $M$ is only $O(\frac{1}{1-M/K})\sqrt{\log(K)}$. This indicates that the regret caused by delay does not increase rapidly as $K$ and $M$ increase.
> > > > >
> > > > > We hope these clarifications address your questions and provide further insight into the results. Please feel free to reach out if you have any additional questions.

---

> > > > > > ### Comment · Reviewer_ix9S · 2024-11-27
> > > > > >
> > > > > > Thanks for answering my questions. However, it does not answer my question 1. Perhaps, I should have clarified before. I get the definition of regret which is external regret. My question was whether the bound achieved in Theorem 2 was achieved with high probability. I mean, can we say that with probability $1-\delta$, $R_T$ is upper bounded by...?

---

> > > > > > > ### Author Response · Authors · 2024-12-03
> > > > > > >
> > > > > > > Thank you for pointing this out and for allowing me to clarify further.
> > > > > > >
> > > > > > > The key lemma in proving Theorem 2 is Lemma 6, which provides the probability of incorrect delay estimation. It should be noted that $\sigma _ d$ appears in the denominator and could potentially be large, which means the bound in Theorem 2 does not always hold with high probability. However, $n$ in Lemma 6 is actually the number of samples on delay. As $t$ increases, $n$ can be very large. This increasing $n$ can balance the impact of a large $\sigma _ d$.
> > > > > > >
> > > > > > > We will improve the proof of Theorem 2. As OpenReview does not allow updates to the PDF now, we briefly explain the improvement here. The expected regret should include a term related to incorrect delay estimation. This term is the probability in Lemma 6 multiplied by $T$. Note that after multiplying by $T$, the regret of incorrect $\hat{\sigma} _ d$ (We take $\hat{\sigma} _ d$ as an example because the regret of incorrect $\hat{\mu} _ d$ is less) is
> > > > > > >
> > > > > > > $$
> > > > > > > {\frac{1}{T}}^{\left[ \frac{nK^2M^2}{320\sqrt{2}\sigma} - 1 \right]}
> > > > > > > $$
> > > > > > >
> > > > > > > We consider two situations: $n \geq \frac{320\sqrt{2}\sigma _ d}{K^2M^2}$ and $n < \frac{320\sqrt{2}\sigma _ d}{K^2M^2}$. When $n \geq \frac{320\sqrt{2}\sigma _ d}{K^2M^2}$, this regret is directly bounded by $1$. Otherwise, we bound the regret using the number of times players select arms. Due to delay, the number of times players select arms is not the same as $n$, but they will have a relation by applying Lemma 4. Thus, the regret when $n < \frac{320\sqrt{2}\sigma _ d}{K^2M^2}$ should be $\frac{640\sqrt{2}\sigma _ d}{\theta K^2M^2} + d(\theta)$, which does not affect the near-optimality of our result. We will check it carefully and update our proof.

---

### Official Review · Reviewer_N47W · 2024-11-03

**Soundness:** 2
**Presentation:** 1
**Contribution:** 2
**Rating:** 3
**Confidence:** 4

**Summary:**

The paper studies the multi-player, multi-armed bandit problem. The difference from the studies is that the authors allow the feedback to be received with a random delay.
The authors develop an algorithm named DDSE and upper bound its performance. They establish that the algorithm is near optimal by deriving a lower bound.

**Strengths:**

The paper provides a detailed analysis of the algorithms and establishes a low bound. However, I could not verify all the claims due to presentation issues.

**Weaknesses:**

The authors consider the multi-player multi-armed bandit problem with a leader-follower structure. Several authors explore this problem. The new dimension of delayed feedback is a minor extension. In addition, I have concerns about the following aspects:

1. The literature review is not detailed: Several papers consider multi-player bandits with a more general heterogenous reward structure, which is well suited for cognitive radio networks.
2. The algorithm is hard to understand: (see details below)
3. The experiments section is weak: Why only compare with SIC-MMAB and not with other algorithms like Game-of-Thrones and Explore-Signal-Exploit Repeat

**Questions:**

It is hard to understand the DDSE algorithm.

1.  What is the duration of exploration, communication, and exploitation?
2. Line 190 says, "the best empirical set arm set of player j." How is this set defined?
3. Line 204: "To avoid collision with followers and ensure sufficient exploration, the leader first sequentially hops in the set of best empirical arms with followers." How is it ensured that the best empirical arm of leader and follower do not overlap? How is the collision avoided?
4. How are collisions interpreted in the communication phase? Is it binary signaling?


In the experiment section, why are the algorithms in any of the following papers not considered?
1. http://proceedings.mlr.press/v83/besson18a/besson18a.pdf
2. http://papers.neurips.cc/paper/7952-distributed-multi-player-bandits-a-game-of-thrones-approach.pdf
3. https://ieeexplore.ieee.org/stamp/stamp.jsp?arnumber=8737653

Minor issues:

1. Line 209: s<T or s<t?
2. Is there any difference between sequential hopping and round-robin?
3. Notation say [n]={1,2,..,n}. Then why |[K]| is M, not K?

---

> ### Author Response · Authors · 2024-11-22
> **Response to Reviewer N47W (Part 1)**
>
> Thank you for your comments about our manuscript. We have studied the comments carefully, and find them valuable for improving our paper. The responses to your comments are as follows.
> # Response to Weakness 1:
> *Q: The literature review is not detailed.*
>
> Thank you for your valuable feedback. We have expanded the literature review to include a discussion of several papers that address multi-player bandits with more general heterogeneous rewards. Please refer to Appendix A for the updated related works section.
>
> # Response to Question 1:
> *Q: What is the duration of exploration, communication, and exploitation?*
>
> The length of a communication phase is $K+2M$. The total duration of exploration is in Equation (14). Players are in exploration phase. They enter a communication phase every $KM\log(T)$ times. After the leader eliminates all sub-optimal arms and communicates the ending signal of exploration in the next communication phase, She begins the exploitation phase and continuously selects her arm until $T$. As for followers, they select arms in $\mathcal{M}^j _ {p-q _ j}$ and enter the communication phase every $KM\log(T)$ to receive the collision from the leader. After they receive a collision from the third block from a certain communication phase, they save the number of the final communication phase as $p _ {\max}$. Then followers will no longer enter a communication phase at the left time. All they need to do is select arms in $\mathcal{M} _ {p _ {\max}-q _ j}$. With time going by, $q _ j=0$ and followers select arms in the final best arm set, meaning that they are in exploitation.
>
> # Response to Question 2:
> *Q: Line 190 says, "the best empirical set arm set of player j." How is this set defined?*
>
> We define $\mathcal{M} _ {p}^j$ as the best empirical arm set of player $j$ at phase $p$. In the beginning, the set is randomly initialized. As the leader explores, she gets empirical rewards of arms and ranks these arms with their rewards. The $M$-th first should be put into the set. After finding an update, the leader passes the information to followers in the communication phase.
>
> # Response to Question 3:
> *Q: How is it ensured that the best empirical arm of leader and follower do not overlap in the exploration phase? How is the collision avoided?*
>
> To ensure sufficient exploration, the leader should explore all arms. However, followers are selecting arms in $\mathcal{M} _ {p-q _ j}^j$ in a round-robin way. Thus, we have a sophisticated design for leader's arm selection. We first give an example with $K=8$ and $M=4$:
>
> \begin{aligned}
> \mathrm{Leader:} \ &\mathcal{M}^4_1 = \\{4, 2, 6, 3\\} \quad \mathrm{to} \quad \mathcal{M}^4_2 = \\{4, 2, 6, 8\\} \\\\
> \mathrm{Follower\ 1:} \ &\mathcal{M}^1_1 = \\{4, 2, 6, 3\\} \\\\
> \mathrm{Follower\ 2:} \ &\mathcal{M}^2_1 = \\{4, 2, 6, 3\\} \\\\
> \mathrm{Follower\ 3:} \ &\mathcal{M}^3_1 = \\{4, 2, 6, 3\\}
> \end{aligned}
>
> In this example, we suppose that they are focusing on $\mathcal{M} _ 1^j$. They select arms as follows:
>
> | Player     | $t_1$ | $t_2$ | $t_3$ | $t_4$ | $t_5$ | $t_6$ | $t_7$ | $t_8$ |
> | ---------- | ----- | ----- | ----- | ----- | ----- | ----- | ----- | ----- |
> | Leader     | 3     | 4     | 2     | 6     | 1     | 5     | 7     | 8     |
> | Follower 1 | 4     | 2     | 6     | 3     | 4     | 2     | 6     | 3     |
> | Follower 2 | 2     | 6     | 3     | 4     | 2     | 6     | 3     | 4     |
> | Follower 3 | 6     | 3     | 4     | 2     | 6     | 3     | 4     | 2     |
>
> The leader first selects arms in $\mathcal{M} _ 1^4$ with followers. After pulling all arms in $\mathcal{M} _ 1^4$, she begins to select other arms in $\mathcal{K}\backslash \mathcal{M} _ 1^4$. (Here $\mathcal{K}$ denotes the active arm set, to avoid the abuse of $|[K]|$.) The process continues for $KM\log(T)$ rounds.
>
> In our supplemental material, you can find the related code in  "$\mathtt{ReviewCode/ours/ddse.py}$". We have implemented the process of arm selection in the function "$\mathtt{play()}$".

---

> ### Author Response · Authors · 2024-11-22
> **Response to Reviewer N47W (Part 2)**
>
> # Response to Question 4:
> *Q: How are collisions interpreted in the communication phase? Is it binary signaling?*
>
> The collisions are just collisions that represent that at least two players select the same arm at the time. We design the communication phase to handle the difficulty that players can not directly communicate with others. Although there is no way to pass the information directly under the environment, our algorithms build a specific framework for players to pass information in such an environment. Via the algorithm, collisions can be interpreted as some information by players.
>
> Next, we explain our communication phase here. The primary goal of communication phase is to pass the update of $\mathcal{M} _ p^j$.  Each communication phase is composed of three blocks. We consider $M=4$ and $K=6$. The example is:
>
> \begin{aligned}
> \mathrm{Leader:} \ &\mathcal{M}^4_1 = \\{4, 2, 6, 3\\}, \mathcal{M}^4_2 = \\{4, 2, 6, 1\\}, \mathcal{M}^4_3 = \\{4, 2, 6, 5\\} \\\\
> \mathrm{Follower\ 1:} \ &\mathcal{M}^1_1 = \\{4, 2, 6, 3\\} \\\\
> \mathrm{Follower\ 2:} \ &\mathcal{M}^2_1 = \\{4, 2, 6, 3\\} \\\\
> \mathrm{Follower\ 3:} \ &\mathcal{M}^3_1 = \\{4, 2, 6, 3\\}
> \end{aligned}
>
> ## The first block is used for removing an arm from $\mathcal{M}^j _ p$ .
> The leader first finds the bad arm $a^- _ p$ in $\mathcal{M}^j _ p$ and identifies the position of $a^- _ p$ . Then the leader selects the $i_{a^- _ p}$ -th arm in $\mathcal{M}^j _ {p-q _ j}$ for $M$ times. Followers still select arms in $\mathcal{M}^j _ {p-q _ j}$ in a round-robin way. After a collision occurs, followers save the position that a collision happens. Then when she gets both the position of the arm to be removed and the arm to be added, she will update the best arm set.
> In the example, players in phase $2$ (i.e., $p=2$) but they still use $\mathcal{M} _ 1^j$. $t _ 1$ to $t _ 4$ denotes the rounds in the first block. We suppose that the leader wants to remove arm $1$ from $\mathcal{M} _ 2^4$. The arm selection is in the table.
>
> |            | $t _ 1$ | $t _ 2$ | $t _ 3$ | $t _ 4$ |
> | ---------- | ----- | ----- | ----- | ----- |
> | Leader     | 3     | 3     | 3     | 3     |
> | Follower 1 | 4     | 2     | 6     | 3     |
> | Follower 2 | 2     | 6     | 3     | 4     |
> | Follower 3 | 6     | 3     | 4     | 2     |
>
> Each follower collides on arm $3$ once. They remember the position of collision in $p=2$ is $4$.
> ## The second block is used for adding an arm from $\mathcal{M}^j _ p$.
> The leader finds a better arm that has higher empirical rewards but not in $\mathcal{M}^j _ p$. Then she wants to pass the new arm to followers. Because the new arm $a^+ _ p$ might not be in $\mathcal{M}^j _ p$ or $\mathcal{M}^j _ {p-q _ j}$, we can not let followers receive the collision information via the best arm set. Thus, we utilize the whole arm set. The leader continuously selects $a^+ _ p$ for $K$ times. Followers select all arms in a round-robin way. This block continues $K$ times because the length of the original arm is $K$ and we hope to use the original whole arm set to pass the information.
>
> In the example, $t _ 5$ to $t _ 6$ denotes the rounds in the second block. We suppose that the leader wants to add arm $1$.
>
> |            | $t_5$ | $t_6$ | $t_7$ | $t_8$ | $t_9$ | $t_{10}$ |
> | ---------- | ----- | ----- | ----- | ----- | ----- | -------- |
> | Leader     | 5     | 5     | 5     | 5     | 5     | 5        |
> | Follower 1 | 1     | 2     | 3     | 4     | 5     | 6        |
> | Follower 2 | 2     | 3     | 4     | 5     | 6     | 1        |
> | Follower 3 | 3     | 4     | 5     | 6     | 1     | 2        |
>
> Each follower collides on arm $5$ once. They remember that the collision occurred on arm 5.
> ## The third block is used for passing the ending of exploration.
> Leader explores and gradually eliminates all sub-optimal arms from $[K]$. However, as for followers, they do not know when the exploration of the leader ends. If the leader's exploration has ended and passes all optimal arms to followers, continuing to enter the communication phase is a waste of time for followers. Therefore, a block used for passing the ending of exploration is necessary. Players' actions are similar to the first block.
> ## After a period of delay:
> Note that the collisions in the first and second blocks can not be received immediately. After a period of delay, follower $j$ receives the position $4$ and the arm to be added is $5$. She also knows the feedback is from phase $2$. Therefore, she updates:
> $$
> \mathcal{M}_1^j = \\{4, 2, 6, 3\\} \rightarrow \mathrm{replace\ the\ arm\ in\ position\ 4\ to\ arm\ 5} \rightarrow\mathcal{M}^j_2 = \\{4, 2, 6, 5\\}
> $$
> No matter how late the feedback is received by the follower, she can update the correct $\mathcal{M} _ p^j$.

---

> ### Author Response · Authors · 2024-11-22
> **Response to Reviewer N47W (Part 3)**
>
> # Response to Question on Experiments:
> *Q: In the experiment section, why are the algorithms in [1, 2, 3] papers not considered?*
>
> Thank you for your question. We have included comparisons with the algorithms from these papers. The experimental results are presented in Section 5 and detailed further in Appendix B. Our algorithms demonstrate superior performance compared to these methods. Additionally, we discuss the results and analyze why these algorithms do not perform well on Page 9.
>
> Specifically, [1] design a special UCB index which decreases when collision occurs. However, as the name suggests, players in this algorithm are selfish and only want to maximize their own rewards. Thus, they fail to utilize the exploration results of others, causing the regret to increase rapidly as $M$ grows. Both Game of Throne [2] and ESER [3] follow an explore-then-commit approach, so they rely on the adjustment of parameters heavily. Meanwhile, MCTopM and RandomTopM from [1] are built on the Musical Chair framework [4], where players randomly preempt a chair with no collision. When delay happens, an arm that is identified to be idle in earlier rounds may already have been preempted by other players, but the player always gets out-of-date feedback, resulting in non-stop exploration to find idle arms.
>
> More importantly, none of these algorithms are involved in simulating communication by collisions. The flow of information between players is helpful for them to find optimal arms and lead that the main term in the regret bound is not multiplied by $M$. Our experimental results in Figure 2 also show that the regret of [1, 2, 3] grows rapidly when $M$ increases, while our algorithms are more stable.
>
> As Reviewer fYEh suggested, we also add experiments on a real-world dataset following the work of [5]. See our update on Page 10. The dataset can be found at https://zenodo.org/records/1293283. The throughput is computed using Shannon Formula which is also aligned with [5]. We also compare the cumulative collisions in this experiment, following the experiment in [5, 6]. The results demonstrate that our algorithms outperform the others.
>
> We observe that the regrets of some algorithms that in our comparison increase rapidly, so we evaluated DDSE in both decentralized and centralized settings as Reviewer j5FD suggested. Experimental results in Figure 10 show that the performance of DDSE in decentralized setting closely matches that in centralized setting.
>
> # Response to Minor Issues:
> *Q: (1) Line 209: $s<T$ or $s<t$. (2) Is there any difference between sequential hopping and round-robin? (3) Why $|[K]|=M$ ?*
>
> 1. Thank you for pointing this out. We have corrected it to $t$.
>
> 2. We sincerely appreciate your observation. The terms "sequential hopping" and "round-robin" indeed refer to the same concept. To avoid confusion, we have updated the paper to consistently use "round-robin" throughout our paper.
>
> 3. We are truly grateful for your attention to the notation. Our previous notation was slightly ambiguous.  As explained in "Respond to Question 4", in multi-player bandits, each player has her optimal arm so we totally have $M$ optimal arms. When the leader eliminates all sub-optimal arms, i.e., there are $M$ optimal arms left, she will send collisions in the third block of the next communication phase and then begin exploitation. To clarify, we have introduced an active arm set $\mathcal{K}$ and replaced instances of $[K]$ with $\mathcal{K}$ where appropriate. This ensures the notation aligns with the context more accurately.
>
> # Reference
> [1] Besson, Lilian, and Emilie Kaufmann. "Multi-player bandits revisited." _Algorithmic Learning Theory_. PMLR, 2018.
>
> [2] Bistritz, Ilai, and Amir Leshem. "Distributed multi-player bandits-a game of thrones approach." _Advances in Neural Information Processing Systems_ 31 (2018).
>
> [3] Tibrewal, Harshvardhan, et al. "Distributed learning and optimal assignment in multiplayer heterogeneous networks." _IEEE INFOCOM 2019-IEEE Conference on Computer Communications_. IEEE, 2019.
>
> [4] Rosenski, Jonathan, Ohad Shamir, and Liran Szlak. "Multi-player bandits–a musical chairs approach." _International Conference on Machine Learning_. PMLR, 2016.
>
> [5] Alipour-Fanid, Amir, et al. "Multiuser scheduling in centralized cognitive radio networks: A multi-armed bandit approach." _IEEE Transactions on Cognitive Communications and Networking_ 8.2 (2022): 1074-1091.
>
> [6] Wang, Wenbo, et al. "Decentralized learning for channel allocation in IoT networks over unlicensed bandwidth as a contextual multi-player multi-armed bandit game." _IEEE Transactions on Wireless Communications_ 21.5 (2021): 3162-3178.

---

> > ### Comment · Reviewer_N47W · 2024-11-24
> > **Concerns with experiments**
> >
> > The updates in the experimental section do not convince me. The statement "More importantly, none of these algorithms are involved in simulating communication by collisions" is not true. I will keep my current score.

---

> ### Author Response · Authors · 2024-11-25
> **Response to Concerns with experiments**
>
> "None of these algorithms are involved in simulating communication by collisions" means that [1, 2, 3] do not design a specific communication phase where players pass the reward or arm choice to others. Players in [1, 2, 3] do not utilize the exploration results of others. Therefore, the regrets in [1, 2, 3] are multiplied by $M$. Our regret of DDSE is $O(\log(T))$ due to the communication phase where players communicate on the update of $\mathcal{M} _ {p}^j$.
>
>
> |                Algorithms                |                    Regret                        |
> | ------------------------------ | ------------------------------------------ |
> | Game of Throne [1]                 | $O(M\log^{2+\delta}(T))$                   |
> | MCTopM-kl_UCB [2]                  | $G _{M,\mathbf{\mu}}\log(T)$                |
> | Selfish [2]                        | unknown                                    |
> | ESER (known $\Delta _{\min}$) [3]   | $O(M^2K\log(T))$                           |
> | ESER (unknown $\Delta _{\min}$) [3] | $O(M^2K\Delta _{\max}\log(T)^{1+\beta}(T))$ |
> | Ours                           | $O(\log(T))$                               |
>
> [1, 2, 3] only do experiments on small $K$ and $M$. When $M$ increases, the results are bad. In our experiments, we do experiments with at most $K=50$ and $M=40$.
>
>
> |     | Experiment parameter      |
> | --- | ------------------------- |
> | [1] | $K=M=5$                   |
> | [2] | at most $K=9$ and $M=6$   |
> | [3] | $K=12$, $M=\{6, 10, 12\}$ |
>
> We originally did not compare with [1, 2, 3] because they do not have the communication phase. In contrast, comparison with SIC-MMAB [4] is more suitable.  Experimental results in Figure 10 have already shown that performance of DDSE in decentralized setting closely matches that in the centralized setting. We do not know why Reviewer N47W asks us to compare with [1, 2, 3] and is not satisfied with the comparison results. However, we would like to explain the experiments if you still have questions.
>
> [1] Bistritz, Ilai, and Amir Leshem. "Distributed multi-player bandits-a game of thrones approach." _Advances in Neural Information
>
> [2] Besson, Lilian, and Emilie Kaufmann. "Multi-player bandits revisited." _Algorithmic Learning Theory_. PMLR, 2018.
>
> [3] Tibrewal, Harshvardhan, et al. "Distributed learning and optimal assignment in multiplayer heterogeneous networks." _IEEE INFOCOM 2019-IEEE Conference on Computer Communications_. IEEE, 2019.
>
> [4] Boursier, Etienne, and Vianney Perchet. "SIC-MMAB: Synchronisation involves communication in multiplayer multi-armed bandits." Advances in Neural Information Processing Systems 32 (2019).

---

> > ### Comment · Reviewer_N47W · 2024-11-25
> > **communication by collisions**
> >
> > The algorithm in [2] does have a communication phase, which is referred to as the signaling phase. In this phase, the players share their reward information with others through collisions so that everyone has the same reward information at the end of this pahse. Hence the statement  "[1, 2, 3] because they do not have the communication phase" is incorrect.

---

> ### Author Response · Authors · 2024-11-26
> **Response to Reviewer N47W**
>
> Thanks for your valuable feedback. We recognize that [1] includes a signaling phase, and we acknowledge that there were inaccuracies in our initial explanation of this work. However, we would like to emphasize that our experiments were conducted rigorously and correctly. This algorithm was implemented using an open-source GitHub library [2]. We have updated our supplementary material since 22 Nov 2024. For further clarification, we invite you to review the code provided in "$\texttt{ReviewCode/tibrewal2019}$". It should be noted that ESER performs well when parameters are appropriate (e.g. Figure 2 (a)), but its regret grows rapidly when $K$ and $M$ increase.
>
> [1] Tibrewal, Harshvardhan, et al. "Distributed learning and optimal assignment in multiplayer heterogeneous networks." _IEEE INFOCOM 2019-IEEE Conference on Computer Communications_. IEEE, 2019.
>
> [2] Wang, Wenbo, et al. "Decentralized learning for channel allocation in IoT networks over unlicensed bandwidth as a contextual multi-player multi-armed bandit game." _IEEE Transactions on Wireless Communications_ 21.5 (2021): 3162-3178. https://github.com/wbwang2020/MP-MAB

---

### Official Review · Reviewer_WghT · 2024-11-04

**Soundness:** 2
**Presentation:** 1
**Contribution:** 2
**Rating:** 3
**Confidence:** 4

**Summary:**

Multi-player multi-armed bandits have been researched for a long time due to their application in cognitive radio networks. In this setting, multiple players select arms at each time and instantly receive the feedback. Most research on this problem focuses on the content of the immediate feedback, whether it includes both the reward and collision information or the reward alone. However, delay is common in cognitive networks when users perform spectrum sensing. This paper designs a decentralized delayed successive elimination (DDSE) algorithm in multi-player multi-armed bandits with stochastic delay feedback and establish a regret bound. This algorithm enables players to adapt to delayed feedback and avoid collision.

**Strengths:**

In order to address the challenge of delay in cognitive radio networks, this paper proposes a novel bandit framework where multiple players engage in a multi-armed bandit and if two or more players select the same arm, none of them receive the reward. In this framework, players receive feedback after a period of stochastic delay, which complicates their ability to learn and adapt in real time, making it exceedingly difficult to avoid collisions and optimize performance. To solve this problem, this paper designs a DDSE algorithm in multi-player multi-armed bandits with stochastic delay feedback and establish a regret bound of the proposed algorithm.

**Weaknesses:**

The paper provides a series of technical results,  but the writing is muddled and many key symbols are not explained. It is very hard to get some intuition about the approach and its possible advantages and disadvantages. In addition, the description of the algorithm is full of confusion, with many unexplained symbols inside.

1. Why set the length of each communication phase as $K+2M$? The authors should explain the reasons for the design. If the length of communication phase becomes time-varying, will the methods in this paper still apply?

2. The paper provides an analysis of the lower bound for centralized algorithm in Theorem 3, but lacks an analysis of the lower bound for decentralized algorithm, which should be the main focus of the paper.

3. According to Theorem 1 and Theorem 2, DDSE has better convergence performance than DDSE without delay estimation. However, in larger scenarios (Fig. 4(d)), DDSE without delay estimation performs better than DDSE. What is the significance of considering delay estimation in delay estimation algorithms in large-scale scenarios?

4. This paper lacks a description of the proof process for the theorems. In addition, the result of Theorem 1 is complex and the paper lacks specific explanations for these terms.

**Questions:**

1. The writing is confused, many symbols are written incorrectly, and there are symbols that are not explained. For example,
(1) On line 157 of page 3, $r^{j}(s)$ should be written as $r^{j}_{k}(s)$; $\mu_{k}$;
(2) what is the difference between $\mu_{k}$ and $\mu_{(k)}$;
(3) The definition of $N_{t}(k)$ on line 210 of page 4 is error;
(4) The $\mathcal{M}_{0}$ in Algorithm 1 should be  $\mathcal{M}^{M}_{0}$;
(5) What is the $\mathcal{M}_{com}$ in Algorithm 1?

2. The introduction of the Algorithm 1 is very confusing. For example,
(1) What does the line 10 line of the Algorithm 1?
(2) In the model, author claim that $M\leq K$, but in the Algorithm 1, $|[K]|=M$ is used as a criterion for judgment. Please explain this issue.

---

> ### Author Response · Authors · 2024-11-22
> **Response to Reviewer WghT (Part 1)**
>
> Thank you for your comments about our manuscript. We have studied the comments carefully, and find them valuable for improving our paper. The responses to your comments are as follows.
> # Response to Weakness 1:
> *Q: Why set the length of each communication phase as $K+2M$? If the length of communication phase becomes time-varying, will the methods in this paper still apply?*
>
> Each communication phase is composed of three blocks.
> ## The first block is used for removing an arm from $\mathcal{M}^j _ p$ .
> We first explain the case that players do not delay the update of $\mathcal{M}^j _ p$. The leader continuously selects the arm to be removed for $M$ times. Followers selects arms in $\mathcal{M}^j _ {p}$. Because the length of $\mathcal{M}^j _ p$ is $M$, during the process of round-robin selection, each follower will collide with the leader once. The arm that generates a collision will be removed from $\mathcal{M}^j _ p$.
>
> If the update of $\mathcal{M}^j _ p$ is delayed, i.e. players should use $\mathcal{M}^j _ {p-q _ j}$ instead of $\mathcal{M}^j _ {p}$. During this block, we still hope to pass the information that **an arm in $\mathcal{M}^j _ p$ should be removed.** Therefore, the leader firstly finds the bad arm $a^- _ p$ in $\mathcal{M}^j _ p$. If the leader directly selects $a _ p^-$, followers will get the wrong information because they are selecting arms in $\mathcal{M}^j _ {p-q _ j}$ which is not the same with $\mathcal{M}^j _ p$. We consider the example with $M=4$ and $K=6$:
>
> \begin{aligned}
> \mathrm{Leader:} \ &\mathcal{M}^4_1 = \\{4, 2, 6, 3\\}, \mathcal{M}^4_2 = \\{4, 2, 6, 5\\} , \mathcal{M}^4_3 = \\{4, 2, 6, 1\\} \\\\
> \mathrm{Follower\ 1:} \ &\mathcal{M}^1_1 = \\{4, 2, 6, 3\\} \\\\
> \mathrm{Follower\ 2:} \ &\mathcal{M}^2_1 = \\{4, 2, 6, 3\\} \\\\
> \mathrm{Follower\ 3:} \ &\mathcal{M}^3_1 = \\{4, 2, 6, 3\\}
> \end{aligned}
>
> In this example, the leader wants to remove arm $5$ from $\mathcal{M}^4 _ 2$. Note that followers do not know $\mathcal{M}^j _ 2$ and can only select arms in $\mathcal{M}^j _ 1$ in the communication phase. If the leader simply selects arm $5$ in this communication phase, followers will not receive the collision because they just select arm ${4,2,6,3}$. This will lead to missing information from the communication phase.
>
> To avoid this situation, the leader also finds the position of $a^- _ p$ . We define the position of $a^- _ p$ as $i _ {a^- _ p}$. Then the leader selects the $i _ {a^- _ p}$ -th arm in $\mathcal{M}^j _ {p-q _ j}$ for $M$ times. Followers still select arms in $\mathcal{M}^j _ {p-q _ j}$ in a round-robin way. After a collision occurs, followers save the position that a collision happens. Then when she gets both the position of the arm to be removed and the arm to be added, she will update the best arm set. In this example, the leader finds that the position of arm $5$ is $4$, so she selects the $4$-th arm in $\mathcal{M}^4 _ 1$, i.e. arm $3$, for $M$ times. Followers select arm $\{4,2,6,3\}$ in a round-robin way. The detailed arm selection is in the table.
>
> |            | $t _ 1$ | $t _ 2$ | $t _ 3$ | $t _ 4$ |
> | ---------- | ----- | ----- | ----- | ----- |
> | Leader     | 3     | 3     | 3     | 3     |
> | Follower 1 | 4     | 2     | 6     | 3     |
> | Follower 2 | 2     | 6     | 3     | 4     |
> | Follower 3 | 6     | 3     | 4     | 2     |
>
> Each follower collides on arm $3$ once. They remember the position of collision in $p=2$ is $4$.
>
> ## The second block is used for adding an arm from $\mathcal{M}^j _ p$.
> The leader finds a better arm that has higher empirical rewards but not in $\mathcal{M}^j _ p$. Then she wants to pass the new arm to followers. Because the new arm $a^+ _ p$ might not be in $\mathcal{M}^j _ p$ or $\mathcal{M}^j _ {p-q _ j}$, we can not let followers receive the collision information via the best arm set. Thus, we utilize the whole arm set. The leader continuously selects $a^+ _ p$ for $K$ times. Followers select all arms in a round-robin way. This block continues $K$ times because the length of the original arm is $K$ and we hope to use the original whole arm set to pass the information.
> In the example, $t _ 5$ to $t _ 6$ denotes the rounds in the second block. We suppose that the leader wants to add arm $1$.
>
> |            | $t _  5$ | $t _ 6$ | $t _ 7$ | $t _ 8$ | $t _ 9$ | $t _ {10}$ |
> | ---------- | ----- | ----- | ----- | ----- | ----- | -------- |
> | Leader     | 5     | 5     | 5     | 5     | 5     | 5        |
> | Follower 1 | 1     | 2     | 3     | 4     | 5     | 6        |
> | Follower 2 | 2     | 3     | 4     | 5     | 6     | 1        |
> | Follower 3 | 3     | 4     | 5     | 6     | 1     | 2        |
>
> Each follower collides on arm $5$ once. They remember that the collision happens on arm $5$.

---

> ### Author Response · Authors · 2024-11-22
> **Response to Reviewer WghT (Part 2)**
>
> ## The third block is used for passing the ending of exploration.
> Leader explores and gradually eliminates all sub-optimal arms from $[K]$. However, as for followers, they do not know when the exploration of the leader ends. If the leader's exploration has ended and passes all optimal arms to followers, continuing to enter the communication phase is a waste of time for followers. Therefore, a block used for passing the ending of exploration is necessary.
>
> In this block, we can utilize $\mathcal{M}^j _ {p-q _ j}$ to pass information. The length of $\mathcal{M}^j _ {p-q _ j}$ is $M$, so this block continues for $M$ times. In multi-player bandits, each player has her optimal arm so we have $M$ optimal arms in total. When the leader eliminates all sub-optimal arms, i.e., there are $M$ optimal arms left, she will send collisions in the third block of the next communication phase. Otherwise, she does not send collisions in this block.
>
> ## After a period of delay:
> Note that the collisions in the first and second blocks can not be received immediately. After a period of delay, follower $j$ receives the position $4$ and the arm to be added is $5$. She also knows the feedback is from phase $2$. Therefore, she updates:
> $$
> \mathcal{M} _ 1^j = \\{4, 2, 6, 3\\} \rightarrow \mathrm{replace\ the\ arm\ in\ position\ 4\ to\ arm\ 5} \rightarrow\mathcal{M}^j_2 = \\{4, 2, 6, 5\\} \\
> $$
> Therefore, although players use the previous arm set $\mathcal{M} _ 1^j$, followers can still receive the correct update information of $\mathcal{M}_2^j$ even though they do not know $\mathcal{M} _ 2^j$.
>
> In summary, the length of $K+2M$ is enough short to pass the information between players. There is no need to design a time-varying length.
>
> # Response to Weakness 2:
> *Q: The paper lacks an analysis of the lower bound for decentralized algorithm, which should be the main focus of the paper.*
>
> In decentralized multi-player bandits, players intentionally collide with others to simulate communication, which inevitably results in some regret. Therefore, our goal is to minimize the communication duration and the associated regret. To evaluate this, we compare our results with the centralized lower bound to evaluate how the additional information exchange impacts regret reduction. The goal of decentralized multi-player bandits is to reach the same performance as in the centralized setting. [1, 2, 3] study decentralized multi-player bandits and compare their results with centralized lower bound.
>
> # Response to Weakness 3:
>  *Q: In larger scenarios of experiments, DDSE without delay estimation performs better than DDSE. What is the significance of considering delay estimation in delay estimation algorithms in large-scale scenarios?*
>
> We have updated our experimental results. In all of these experiments, only Figure 8(d) shows that DDSE is slightly worse than DDSE without delay estimation. As the red words explained in Appendix B, the interval of each communication phase is $KM\log(T)$, which is large when $K$ and $M$ increase. A large interval makes sure that followers receive feedback from a communication phase before the next communication phase begins.
>
> However, in large-scale scenarios where the delay is very long, i.e., exceeding $KM\log(T)$, players might update the communication results wrongly so that DDSE without delay estimation has a large fluctuation. The fluctuation is also validated in Figure 9(d) where DDSE without delay estimation has a large standard error. In real-world cognitive networks, we can not know the maximum delay in advance. It is impossible to adjust the interval manually. As nodes in cognitive networks increase, real-world delay is also increasing due to the congestion of the network. It is hard to guarantee that a large $K$ or $M$ is enough to balance the delay.
>
> Therefore, DDSE does not need to adjust the interval of each communication phase. No matter what $K$ or $M$ is, and no matter how large the delay is, DDSE always shows a good performance and has a better guarantee. Experiments also show that DDSE is more stable than DDSE without delay estimation. In real-world applications, DDSE is more robust and can adapt to complicated cognitive networks.

---

> ### Author Response · Authors · 2024-11-22
> **Response to Reviewer WghT (Part 3)**
>
> # Response to Weakness 4:
> *Q: This paper lacks a description of the proof process for the theorems.*
>
> Thanks for your advice. We have added some description in Appendix D. We briefly describe the proof of Theorem 1 here.
> ## Exploration Phase
> In the exploration phase, Lemma 5 ensures that the delayed feedback from the communication phase of all followers is bounded. Then Lemma 6 establishes the accuracy of the estimates for $\mathbb{E}[d]$ and $\sigma _ d^2$. As a result, player $j$ can correctly determine $q _ j$ and align with the same best empirical arm set, thereby preventing collisions caused by inconsistencies between the leader and the followers. Thus, the regret in exploration phase is generated from (1) selecting sub-optimal arms, (2) players not receiving any feedback initially, and (3) the leader not entering the exploitation phase immediately after identifying all sub-optimal arms. During the period after the leader identifies all sub-optimal arms but before entering the exploitation phase, the leader still needs to maintain consistency with followers by selecting arms in $\mathcal{M} _ {p-q _ M}$, i.e., $|\mathcal{K}|=M$, $e _ M=M$ but $q _ M\neq 0$ which do not satisfy Line 14 in Algorithm 1. Then we bound these terms separately in Appendix D.
> ## Communication Phase
> We have already known that the length of each communication phase is $K+2M$. Note that players enter a communication phase every $KM\log(T)$ rounds. The next step is to bound $T_{expl}$ which is the total time that the leader needs to receive feedback and eliminate all sub-optimal arms. Thus, $T _ {expl}/KM\log(T)$ indicates the number of times players enter a communication phase, which we then multiply by the phase length $K+2M$. Note that there are $M$ players in communication phase, so we finally take a union bound of $M$.
>
> # Response to Question 1:
> *Q: The writing is confused, many symbols are written incorrectly.*
>
> We sincerely thank the reviewer for the careful reading and valuable feedback. All noted issues with symbols and writing have been addressed. Below, we provide our detailed responses to the points raised.
> 1. We use the definition $r^j(s)$ because the reward has already been determined by a given player $j$ and a round $s$. There is no need to add a notation of $k$.
> 2. As stated in Line 154, $\mu_  {(k)}$ is the $k$-th highest reward. We have a order $\mu _ {(1)} \geq \mu _ {(2)} \geq ... \geq \mu _ {(K)}$. We define $\mu _ k$ as the expectation of arm $k$.
> 3. Thank you very much for pointing out this one. We have updated it in Line 203. $N_t(k):= \sum_{s \leq t} \mathbb{I}\{\pi_s^j=k, j=M\}$ is the number of times that the leader chooses arm $k$ before $t$.
> 4. We truly appreciate your attention to this detail. The notation has been updated to $\mathcal{M} _ 0^M$.
> 5. Thank you for highlighting this point. $\mathcal{M} _ {com}$ is originally used for determining whether to communicate the update of $\mathcal{M}^j _ p$. If $\mathcal{M}^M _ {p}=\mathcal{M}^M _ {p-1}$, the leader do not need to communicate with followers. We have removed the notation of $\mathcal{M} _ {com}$ and change the if-condition in Algorithms 1 into "$\mathtt{if}\ \mathcal{M}^M _ p \neq \mathcal{M}^M _ {p-1}\ \mathtt{then}$ ".  If $\mathcal{M}^M _ p \neq \mathcal{M}^M _ {p-1}$, the leader send collisions in communication phase. Otherwise, she runs a virtual communication to select arms with followers in $\mathcal{M}^M _ {p-q_M}$ with no collision.
>
> # Response to Question 2:
> *Q: The introduction of the Algorithm 1 is very confusing. e.g. (1) Line 10 line of the Algorithm 1, (2) concern about $|[K]|=M$.*
>
> 1. We have removed the notation of $\mathcal{M} _ {com}$ in Line 10 of Algorithm 1 and use "$\mathtt{if}\ \mathcal{M}^M _ p \neq \mathcal{M}^M _ {p-1}\ \mathtt{then}$ " as the if-else condition. The detailed explanation and example are in "Response to Weakness 1". Thank you for pointing out this improvement.
> 2. The leader explores and gradually eliminates all sub-optimal arms. As stated in "Response to Weakness 1", in multi-player bandits, each player has her optimal arm so we totally have $M$ optimal arms. When the leader eliminates all sub-optimal arms, i.e., there are $M$ optimal arms left, the arm set shrinks into the length of $M$. Therefore, when $[K]$ shrinks to only $M$ elements, the exploration ends.
>
> # Reference
> [1] Boursier, Etienne, and Vianney Perchet. "SIC-MMAB: Synchronisation involves communication in multiplayer multi-armed bandits." _Advances in Neural Information Processing Systems_ 32 (2019).
>
> [2] Shi, Chengshuai, et al. "Decentralized multi-player multi-armed bandits with no collision information." _International Conference on Artificial Intelligence and Statistics_. PMLR, 2020.
>
> [3] Huang, Wei, Richard Combes, and Cindy Trinh. "Towards optimal algorithms for multi-player bandits without collision sensing information." _Conference on Learning Theory_. PMLR, 2022.

---

### Official Review · Reviewer_fYEh · 2024-11-04

**Soundness:** 3
**Presentation:** 3
**Contribution:** 3
**Rating:** 6
**Confidence:** 4

**Summary:**

In this paper, the authors have considered the delayed feedback setting in multi-player multi-armed bandit problem, motivated by cognitive radio applications. A decentralized delayed successive elimination (DDSE) algorithm which takes into account stochastic delay, is proposed in the paper, and a regret bound is established.  Contrary to existing algorithms, the proposed algorithm can avoid collision by adapting to delayed feedback. A corresponding lower bound on the regret is also derived. Experiment results are presented to demonstrate the efficacy of the proposed algorithm.

**Strengths:**

The considered problem is well-motivated and the analysis appears to be sound.

**Weaknesses:**

The proposed algorithm takes a leader-follower approach which makes it semi-distributed in nature as there is a necessity of communication between the leader and the followers. The authors have considered the fixed user setting where no users are allowed to enter or leave the systems. The modeling of delay could have been better.

**Questions:**

My concerns are as follows:
1.	The proposed algorithm takes a leader-follower approach which makes it semi-distributed in nature as there is a necessity of communication between the leader and the followers. There are works in the literature which can work without the requirement of a leader, e.g.,
Trinh, Cindy, and Richard Combes. "A High Performance, Low Complexity Algorithm for Multi-Player Bandits Without Collision Sensing Information." arXiv preprint arXiv:2102.10200 (2021).
2.	What is the rationale behind Assumption 1? What are the components of delay? For example, does it contain queueing delay? How practical is the consideration of sub-Gaussian delay?
3.	The authors have considered the fixed user setting where no users are allowed to enter or leave the systems. However, in a practical cognitive radio application, users may enter or leave the system. How does the proposed algorithm behave when user entering and leaving are allowed in the system?
4.	It is not clear why there is a provision of eliminating arms for which LCB is bigger than UCB. Please specify the motivation behind the virtual communication phase in details.
5.	Please provide a pointer to the result where an upper bound on the feedback delay is derived. This result has been used in Lemma 1.
6.	Can the authors quantify the gap between the lower bound and upper bound on the regret of the proposed algorithm? It will be more justified to call the proposed algorithm near-optimal then.
7.	Since the paper is highly motivated by cognitive radio applications, I expected some real wireless networks simulations (such as ns-3 simulations) where delays will be real delays in a wireless network.

---

> ### Author Response · Authors · 2024-11-22
> **Response to Reviewer fYEh (Part 1)**
>
> Thank you for your comments about our manuscript. We have studied the comments carefully, and find them valuable for improving our paper. The responses to your comments are as follows:
> # Response to Question 1:
> *Q: The proposed algorithm takes a leader-follower approach which makes it semi-distributed.*
>
> We agree that many works on decentralized MMAB do not need to assume that $M$ and the ID of each player are known. Actually, we have considered this fully decentralized setting at the beginning of our work. In such fully decentralized multi-player bandits, players do not know the existence of others so there is no guarantee to avoid collisions. Thus, decentralized MMAB algorithms need an initialization phase where players utilize collisions to get some information before exploration so that they can select arms in a round-robin way. There are two kinds of methods for initialization.
> 1. [1, 2, 3] adopt a musical chair method, where each player preempts an arm until no collision happens. After preempting an arm, players then intentionally select some arms. By counting the times of accumulated collision in this period, each player can know $M$ and her ID among these players.
> 2. In [4], players also try to preempt arms once. If they fail to preempt it, i.e., receive a collision, they will select specific arms later to pass the information that she has not find her arm to other players. When no collision occur, it means that everyone finds the proper arm. Then they perform an procedure similar to [1, 2, 3] and get the information on $ M$ and their ID.
>
> However, when delay is introduced to the environment, players can not receive the collision immediately. If the delay is too long, players must wait in the initialization phase to receive the feedback of collision, because they need to get $M$ and their ID to start an exploration where they can select arms in a round-robin way. Actually, players always hope that they should receive some feedback in a certain known period, which conflicts with the scenario where feedback is delayed for an unknown period.
>
> Of course, if we assume that delay is bounded by a known value $d _ {\max}$, the problem of initialization will be solved easily by using the classical technique and waiting for extra $d _ {\max}$ rounds in the beginning. In cognitive radio networks, transmission delay is sometimes bounded by protocol limits, so it is also natural to consider a bounded delay where we can use $d_{\max}$ as the input of algorithms to perform initialization.
>
> Our assumption on delay is more mild than this bounded delay. We allow some exceedingly large and unknown delays in the environment. If the assumption on the initialization does not exist, i.e., players are fully decentralized, we can also implement that pre-mentioned technique in our algorithms. For example, players start with an initialization phase. Although their feedback might not be completed, they still begin to explore arms. If a player in exploration has a collision that should not have happened, she notifies others also by sending collisions at a specific time. After a period of delay, players receive the signal and start an initialization again. Then comes the next exploration. By doing so, it is possible to give a probability guarantee using our sub-Gaussian delay assumption. However, it will make the notation more complicated. The existing problem is enough challenging and our work is the first paper to handle delay in multi-player bandits where a collision occurs when selecting the same arm. Our goal is to pave the way for further study in this area. A fully decentralized MMAB can be discussed in future work.

---

> ### Author Response · Authors · 2024-11-22
> **Response to Reviewer fYEh (Part 2)**
>
> # Response to Question 2:
> *Q: What is the rationale behind Assumption 1? What are the components of delay? For example, does it contain queueing delay? How practical is the consideration of sub-Gaussian delay?*
> ## Rationale behind Assumption 1
> The rationale behind Assumption 1 lies in providing a realistic yet mathematically tractable model for network delays in multi-player bandit problems. In real-world networks, transmission delays are inherently bounded by physical and protocol limits, which prevent excessively large values. A common assumption on delay is that all delay is bounded by a fixed value $d_{\max}$. However, we choose not to rely on this assumption. Instead, we propose a more flexible sub-Gaussian assumption that permits larger delays but with a low probability of occurrence. It is worth noticing that our assumption is more general than bounded delay, as all bounded random variables are inherently sub-Gaussian. In our analysis, we leverage the property of exponential decay in the tail probabilities characteristic of sub-Gaussian distributions.
> ## Components of delay
> In cognitive radio networks, delay typically consists of several components, depending on the communication process and network architecture. These components can include:
> 1. Propagation Delay: The time required for a signal to travel from the transmitter to the receiver across the communication medium.
> 2. Transmission Delay: The time taken to push all the bits of a packet onto the transmission medium. It depends on the packet size and the bandwidth of the link.
> 3. Processing Delay [5]: The time spent on processing tasks, such as spectrum sensing, packet routing, and other computational operations required before transmission.
> 4. Queueing Delay [6, 7]: The time an SU's data packet spends in a queue, either at the SU’s transmitter or at network devices (e.g., routers or base stations), waiting to be transmitted. Queueing delay occurs due to competition among SUs for limited resources, such as unoccupied PU channels or transmission opportunities.
>
> Thus, queueing delay is an important and common component of the overall delay in cognitive radio networks.
> ## How practical is our assumption
> As stated before,  sub-Gaussian delay is practical and well-suited to the dynamic nature of cognitive radio networks. Sub-Gaussian delay models are characterized by exponential tail decay, which reflects the typical behavior of delays in real-world systems. Most delays are relatively small, but occasionally, larger delays can occur with diminishing probability. This is particularly relevant for cognitive radio networks, where delays may arise from dynamic spectrum access or contention among secondary users for idle channels. Moreover, our assumption is a generalization of bounded delay. Compared with [8, 9, 10], sub-Gaussian delay allows for rare occurrences of larger delays while maintaining tractability. This makes it more flexible and realistic for cognitive radio networks.
>
> # Response to Question 3:
> *Q: Concerns about fixed users in the system.*
>
> If players are not fixed, the setting is called asynchronous multi-player bandits. In this setting, players can enter the game and leave at any time.  The latest study on asynchronous MMAB is [11]. However, [11] relies on several assumptions:
> 1. There exists a centralized environment, where players can freely communicate with others.
> 2. Although players enter the game at different times, they leave at the same time $T$.
> 3. At each time, the probability that every player enters the game is the same and known.
>
> The area of centralized asynchronous MMAB without assumptions is still blank, let alone decentralized asynchronous MMAB.
> In decentralized bandits, the delay has already significantly complicated the problem because players rely on the feedback of collisions to get some information from other players. Incomplete feedback causes inconsistency between players, leading to staggered exploration, frequent collisions, or premature exploitation. We expect to discuss it in the future work.

---

> ### Author Response · Authors · 2024-11-22
> **Response to Reviewer fYEh (Part 3)**
>
> # Response to Question 4:
> *Q: It is not clear why there is a provision of eliminating arms for which LCB is bigger than UCB. Please specify the motivation behind the virtual communication phase in details.*
> ## Arm Elimination
> In multi-player bandits, there are $M$ players and $K$ arms. The ultimate goal is that every player is assigned an optimal arm. Thus, there exist at least $M$ optimal arms. In our algorithms, the leader explores all arms and puts $M$ arms with the $M$-th highest empirical rewards into the best arm set $\mathcal{M} _ p^j$. Therefore, if the potential reward of arm $k$  is worse than that of at least $M$ arms, it should be considered a bad arm because we already have at least $M$ arms that are better than $k$. Then we can eliminate arm $k$. To evaluate how the "potential reward" is, we introduce the LCB and UCB which
> are lower confidence bound and upper confidence bound [12]. They are separately defined as:
> $$
> LCB_t(k):=\hat{\mu}_k(t) - \sqrt{\frac{2\log(T)}{n_t(k)}},\ UCB_t(k):=\hat{\mu}_k(t) + \sqrt{\frac{2\log(T)}{n_t(k)}}.
> $$
> $\hat{\mu}_k(t)$ is the empirical reward expectation of arm $k$ at time $t$ and $n_t(k)$ denotes the number of times that the leader receives feedback on arm $k$ up to $t$. If $UCB_k(t)$ is higher, we deem that arm $k$ is better. In the equation, the higher first term $\hat{\mu}_k(t)$ means higher current rewards, which represent exploitation. Note that $T$ is the total time and $n_t(k)$ indicates how familiar we are with this arm. Higher $T$ and lower $n_t(k)$ indicate that we have less knowledge of arm $k$. Thus, the higher second term $\sqrt{\frac{2\log(T)}{n_t(k)}}$ means the degree of uncertainty, which represent exploration. By Hoeffding's inequality, we have
>
> $$
> P\left(\left|\hat{\mu}_k(t)-\mu_k\right| \leq \sqrt{\frac{2\log(T)}{n_t(k)}}\right) \geq 1-2(\frac{1}{T})^4
> $$
> Thus, by picking an arm with the highest UCB, the player can finally find the optimal solution.
> In our algorithms, if the UCB of arm $k$ is worse than at least $M$ arms' LCB, the arm $k$ will be eliminated because optimal arms exist in the left arms. By eliminating arms gradually, the leader can find $M$ optimal arms in total.
> ## Motivation behind Virtual Communication
> We have modified our algorithms by removing the notation of $\mathcal{M} _ {com}$ as reviewer WghT suggested. Here we briefly describe the communication phase. The leader passes the update of $\mathcal{M} _ p^j$ to followers at each communication phase. In the communication phase, collisions represent removing or adding an arm to $\mathcal{M} _ p^j$. The collision is also used for passing the ending signal of exploration. However, if $\mathcal{M} _ p^j=\mathcal{M} _ {p-1}^j$ and the exploration does not end, it is not necessary for the leader to pass information in the communication phase. From the point of followers, they do not know which communication phase does not have information, so they choose to enter every communication phase between a fixed gap. To maintain alignment with followers, the leader should enter a "virtual communication" phase if she does not want to pass any information. In virtual communication, the leader selects arms with followers and does not send collisions. Later, if followers do not receive any feedback from certain communication phase, they can know that the best arm set $\mathcal{M} _ {p}^j$ is the same as the prior best arm set $\mathcal{M} _ {p-1}^j$.
>
> If virtual communication does not exist, we have the following problems: (1) Because followers always enter a communication phase between a fixed gap, in the second block of communication, they select arms from the whole arm set $[K]$. If the leader does not make adjustments, she might collide with followers. This collision will be regarded by followers as information about adding an arm, but actually, the information is wrong. (2) Followers update $\mathcal{M} _ {p}^j$ based on $\mathcal{M} _ {p-1}^j$, if $\mathcal{M} _ {p-1}^j$ is blank or wrong as (1) mentioned, the updated $\mathcal{M} _ {p}^j$ might also be wrong.
>
> Therefore, virtual communication is critical in our algorithm.
> # Response to Question 5:
> *Q: Please provide a pointer.*
>
> Thank you for your comment. We have updated the paper to include a clear reference to the result where the upper bound on the feedback delay is derived. This result is now explicitly linked to Lemma 1.
>
> # Response to Question 6:
>  *Q: Can the authors quantify the gap between the lower bound and upper bound on the regret of the proposed algorithm?*
>
> Thank you for the suggestion. We have updated the paper to address this point. The first term in Theorem 2 (regret of DDSE) is aligned with Theorem 1 (lower bound) up to constant factors. The observed difference comes from the decentralized setting, where players cannot directly communicate about rewards or collisions. Importantly, the regret introduced by the decentralized structure and delay remains independent of $T$. Therefore, our result is near-optimal.

---

> ### Author Response · Authors · 2024-11-22
> **Response to Reviewer fYEh (Part 4)**
>
> # Response to Question 7:
> *Q: Since the paper is highly motivated by cognitive radio applications, I expected some real wireless networks simulations.*
>
> We have done experiments on a real-world dataset following the work of [13]. See our update on Page 10. The dataset can be found at https://zenodo.org/records/1293283. The throughput is computed using Shannon Formula which is also aligned with [13]. We also compare the cumulative collisions in this experiment, following the experiment in [13, 14]. The results demonstrate that our algorithms outperform the others. However, about ns-3 simulation, we are still in progress and aim to complete this experiment before the deadline.
>
> # Reference
> [1] Boursier, Etienne, and Vianney Perchet. "SIC-MMAB: Synchronisation involves communication in multiplayer multi-armed bandits." _Advances in Neural Information Processing Systems_ 32 (2019).
>
> [2] Shi, Chengshuai, et al. "Decentralized multi-player multi-armed bandits with no collision information." _International Conference on Artificial Intelligence and Statistics_. PMLR, 2020.
>
> [3] Huang, Wei, Richard Combes, and Cindy Trinh. "Towards optimal algorithms for multi-player bandits without collision sensing information." _Conference on Learning Theory_. PMLR, 2022.
>
> [4] Wang, Po-An, et al. "Optimal algorithms for multiplayer multi-armed bandits." _International Conference on Artificial Intelligence and Statistics_. PMLR, 2020.
>
> [5] Ahmad, Wan Siti Halimatul Munirah Wan, et al. "5G technology: Towards dynamic spectrum sharing using cognitive radio networks." _IEEE access_ 8 (2020): 14460-14488.
>
> [6] Wang, Shanshan, Junshan Zhang, and Lang Tong. "Delay analysis for cognitive radio networks with random access: A fluid queue view." _2010 Proceedings IEEE INFOCOM_. IEEE, 2010.
>
> [7] Laourine, Amine, Shiyao Chen, and Lang Tong. "Queuing analysis in multichannel cognitive spectrum access: A large deviation approach." _2010 Proceedings IEEE INFOCOM_. IEEE, 2010.
>
> [8] Li, Yandi, and Jianxiong Guo. "A Modified EXP3 and Its Adaptive Variant in Adversarial Bandits with Multi-User Delayed Feedback." _arXiv preprint arXiv:2310.11188_ (2023).
>
> [9] van der Hoeven, Dirk, et al. "A Unified Analysis of Nonstochastic Delayed Feedback for Combinatorial Semi-Bandits, Linear Bandits, and MDPs." _The Thirty Sixth Annual Conference on Learning Theory_. PMLR, 2023.
>
> [10] Wang, Dairui, et al. "Cascading bandits: optimizing recommendation frequency in delayed feedback environments." _Advances in Neural Information Processing Systems_ 36 (2024).
>
> [11] Richard, Hugo, Etienne Boursier, and Vianney Perchet. "Constant or Logarithmic Regret in Asynchronous Multiplayer Bandits with Limited Communication." _International Conference on Artificial Intelligence and Statistics_. PMLR, 2024.
>
> [12] Lattimore, Tor, and Csaba Szepesvári. _Bandit algorithms_. Cambridge University Press, 2020.
>
> [13] Alipour-Fanid, Amir, et al. "Multiuser scheduling in centralized cognitive radio networks: A multi-armed bandit approach." _IEEE Transactions on Cognitive Communications and Networking_ 8.2 (2022): 1074-1091.
>
> [14] Wang, Wenbo, et al. "Decentralized learning for channel allocation in IoT networks over unlicensed bandwidth as a contextual multi-player multi-armed bandit game." _IEEE Transactions on Wireless Communications_ 21.5 (2021): 3162-3178.

---

> > ### Comment · Reviewer_fYEh · 2024-11-26
> >
> > I thank the authors for their detailed response. Since queuing delay could be arbitrarily large, does the assumption on delays (SubGaussian and generation of bounded delay) still hold?

---

> > > ### Author Response · Authors · 2024-12-03
> > > **Response to Official Comment**
> > >
> > > We acknowledge that in extreme cases with severe congestion, the delays may not adhere to a sub-Gaussian distribution. In such situations, the delay distribution might exhibit heavy-tailed characteristics, and our assumptions would need to be adjusted accordingly. We plan to address this limitation in our revised manuscript by including a discussion on the applicability of our delay assumptions and potential extensions to handle heavy-tailed delay distributions.
> > >
> > > Thank you for your valuable feedback. Your thoughts help us improve the clarity and robustness of our work.

---

### Author Response · Authors · 2024-11-25
**Official Comment by Authors**

We thank you once again for your careful reading of our paper and your constructive comments and suggestions. We will appreciate it very much if you could let us know whether all your concerns are addressed. We are also more than happy to answer any further questions in the remaining discussion period.

---

### Meta-Review · Area_Chair_Xo4z · 2024-12-20

**Metareview:**

The paper addresses the multi-player multi-armed bandit (MP-MAB) problem within cognitive radio networks, where multiple users select channels (arms) and receive immediate feedback. Traditional research in this area typically focuses on the nature of immediate feedback, such as whether it includes both reward and collision information or just the reward. However, in real-world cognitive networks, spectrum sensing often introduces delays in feedback, complicating the decision-making process. To tackle this, the authors propose the Decentralized Delayed Successive Elimination (DDSE) algorithm, specifically designed to handle stochastic delays in feedback. Unlike existing algorithms that do not account for such delays, DDSE enables players to adapt to delayed information and effectively avoid collisions, enhancing overall network performance.

The authors establish a theoretical regret bound for DDSE, demonstrating its efficiency and near-optimal performance by deriving a corresponding lower bound in a centralized setting. This theoretical validation highlights DDSE’s superiority over existing approaches that fail to manage delayed feedback. Additionally, the paper presents comprehensive numerical experiments using both synthetic and real-world datasets, which confirm the algorithm’s effectiveness and practical applicability. Overall, the study makes significant contributions by introducing a robust solution for delayed feedback scenarios in MP-MAB problems, relevant to applications like cognitive radio networks, and by providing both theoretical and empirical evidence of its advantages.

The reviewers' evaluations have wide margins. Two reviewers, specifically, raise questions about the contribution beyond the existing literature, some inconsistent theoretical and empirical observations, and clarity in presentation.

**Additional Comments On Reviewer Discussion:**

Most reviewers and authors engaged in the discussion period. The discussions, however, did not change the verdits of the critical reviewers.

---

### Decision · Program_Chairs · 2025-01-22

Reject